# Generative Model Inversion Through the Lens of the Manifold Hypothesis

**Xiong Peng**[1]  **Bo Han**[1][†]  **Fengfei Yu**[1]  **Tongliang Liu**[2]  **Feng Liu**[3]  **Mingyuan Zhou**[4]

[1]TMLR Group, Department of Computer Science, Hong Kong Baptist University
[2]Sydney AI Centre, The University of Sydney
[3]School of Computing and Information Systems, The University of Melbourne
[4]McCombs School of Business, The University of Texas at Austin

{csxpeng, bhanml}@comp.hkbu.edu.hk    alvinfengfei@gmail.com
tongliang.liu@sydney.edu.au    fengliu.ml@gmail.com    mingyuan.zhou@mccombs.utexas.edu

## Abstract

*Model inversion attacks* (MIAs) aim to reconstruct class-representative samples from trained models. Recent generative MIAs utilize generative adversarial networks to learn image priors that guide the inversion process, yielding reconstructions with high visual quality and strong fidelity to the private training data. To explore the reason behind their effectiveness, we begin by examining the gradients of inversion loss *w.r.t.* synthetic inputs, and find that these gradients are surprisingly noisy. Further analysis reveals that generative inversion implicitly denoises these gradients by projecting them onto the tangent space of the generator manifold, thereby filtering out off-manifold components while preserving informative directions aligned with the manifold. Our empirical measurements show that, in models trained with standard supervision, loss gradients often exhibit large angular deviations from the generator manifold, indicating poor alignment with class-relevant directions. This observation motivates our central hypothesis: models become more vulnerable to MIAs when their loss gradients align more closely with the generator manifold. We validate this hypothesis by designing a novel training objective that explicitly promotes such alignment. Building on this insight, we further introduce a *training-free* approach to enhance gradient-manifold alignment during inversion, leading to consistent improvements over state-of-the-art generative MIAs. The code is publicly available at `https://github.com/tmlr-group/AlignMI`.

## 1 Introduction

Machine learning (ML) models are increasingly deployed in high-stakes domains such as finance [Rundo et al., 2019], healthcare [Richens et al., 2020], and biometrics [Zhong et al., 2024]. Trained on sensitive data, these models are attractive targets for adversarial threats [Fredrikson et al., 2014, Choquette-Choo et al., 2021, Jiang et al., 2023, Zhang et al., 2023]. One emerging threat is the *model inversion attack* (MIA), which exploits model outputs to infer class-sensitive attributes or reconstruct representative samples, thereby posing serious risks to user privacy and security.

Early work by Fredrikson et al. [2015] formulated MIAs as an input-space optimization problem, using gradient descent to find inputs that maximize the prediction score of a target class. This method effectively reconstructed low-resolution grayscale faces from shallow models. However, this approach performs poorly on deep neural networks (DNNs) trained on high-dimensional data (*e.g.,* RGB facial images). Direct optimization in the input space is often ill-posed, because natural images are not

---

[†]Correspondence to Bo Han (bhanml@comp.hkbu.edu.hk).

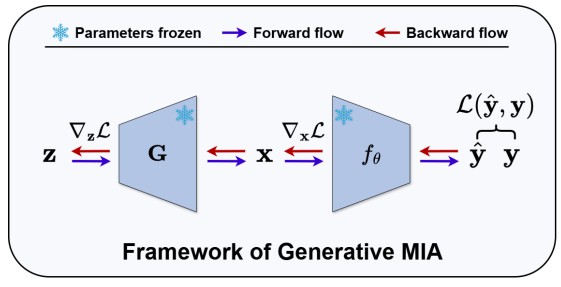

(a) Framework overview

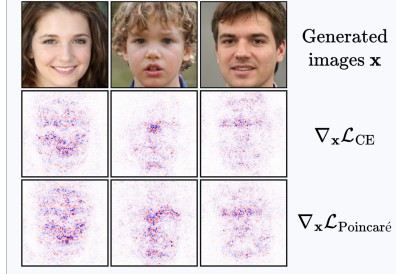

(b) Loss gradient visualization

Figure 1: **Generative MIA framework and loss gradient visualization.** **(a)** Generative model inversion extracts private information via inversion-time classification loss gradients $\nabla_{\mathbf{x}}\mathcal{L}(\hat{y}, y)$ from model $f_{\boldsymbol{\theta}}$. **(b)** Loss gradient visualizations under PPA method in the high-resolution setting, comparing cross-entropy ($\mathcal{L}_{\mathrm{CE}}$) and Poincaré ($\mathcal{L}_{\mathrm{Poincaré}}$) losses. In both cases, the loss gradients appear highly noisy. Additional low-resolution visualizations are provided in Appendix E.4.

uniformly distributed across the input domain, but are instead concentrated on a low-dimensional manifold embedded in a high-dimensional ambient space [Fefferman et al., 2016]. Consequently, reconstructions often fall off the manifold and produce semantically irrelevant features.

To address this challenge, Zhang et al. [2020] introduced the *generative model inversion* framework, which leverages generative adversarial networks (GANs) [Goodfellow et al., 2014, Radford et al., 2016] to learn an image prior from public auxiliary datasets, such as web-scraped facial images. The learned prior constrains the inversion process to the generator's latent space, significantly improving the visual quality and semantic relevance of reconstructed samples. This paradigm has spurred notable progress in the MIA field [Wang et al., 2021a, Struppek et al., 2022, Nguyen et al., 2023b, Peng et al., 2024a], enabling recovery of samples that closely resemble the private training data.

Despite the empirical success of generative MIAs, it remains insufficiently understood how private information encoded in the target model is exploited during the inversion process. To bridge this research gap, we adopt a gradient-based perspective, and begin by closely examining the gradients of the *inversion-time* classification loss *w.r.t.* the synthetic inputs (hereafter referred to as loss gradients) during the inversion optimization process. Surprisingly, we observe that these gradients are highly noisy (see Fig. 1(b)). Based on the manifold hypothesis and a geometric analysis, we show that the generative MIA approach implicitly performs gradient denoising by projecting the loss gradients onto the tangent space of the generator manifold. This projection preserves informative components that lie along the manifold while filtering out noisy directions that deviate from it (see Fig. 2).

To assess how well the loss gradients capture semantically meaningful directions, we measure their alignment with the tangent space of the generator manifold [Bordt et al., 2023, Srinivas et al., 2023]. Specifically, we quantify this alignment by computing the cosine of the angle between the gradient and its projection onto the manifold. Empirical results show that models trained under standard classification supervision (*i.e.,* vanilla models) exhibit consistently low alignment (see Fig. 3), suggesting that their loss gradients often deviate from the generator manifold and therefore encode limited class-relevant information. Motivated by this observation, as well as the intuition that stronger alignment with the generator manifold indicates more informative gradients, we propose the following hypothesis: *Models tend to be more vulnerable to MIAs when their loss gradients are more aligned with the tangent space of the generator manifold.*

To validate this hypothesis, we design a training objective that promotes alignment between loss gradients and the generator manifold during the inversion process. Although this alignment is not directly measurable during training, a key observation bridges the gap: by the chain rule, loss gradients can be expressed as linear combinations of input gradients, *i.e.,* the gradients of the model's outputs *w.r.t.* its inputs. This insight allows us to shift our focus: during training, we can instead encourage alignment between input gradients and the tangent space of the data manifold. To estimate this tangent space, we leverage a pre-trained variational autoencoder (VAE) from Stable Diffusion [Kingma and Welling, 2014, Rombach et al., 2022], which approximates the natural image manifold. Based on this estimate, we propose a novel training objective that augments the standard classification loss with an auxiliary term that promotes input gradients to align with the estimated tangent space.

Building on the empirical validation of our hypothesis, we further propose AlignMI, a training-free approach that enhances gradient-manifold alignment during the model inversion process. The key idea is to sample multiple variants of a synthetic input within a local neighborhood and average their corresponding loss gradients [Smilkov et al., 2017]. This operation attenuates noisy, off-manifold components while amplifying consistent directions aligned with the generator manifold, resulting in a more informative and semantically meaningful gradient signal.

In summary, our contributions are: (1) We present the first geometric analysis of generative model inversion, revealing that it fundamentally operates as an implicit gradient denoising mechanism via projection onto the generator manifold (Sec. 3). (2) This perspective leads us to hypothesize, and empirically validate that stronger gradient-manifold alignment increases a model's vulnerability to MIAs, revealing a previously underexplored dimension of model inversion vulnerability (Sec. 4). (3) Based on this insight, we propose AlignMI, a training-free approach to enhance gradient-manifold alignment during inversion. We instantiate AlignMI with two concrete techniques: *perturbation-averaged alignment* (PAA) and *transformation-averaged alignment* (TAA) (Sec. 5), both of which consistently improve the performance of state-of-the-art (SOTA) generative MIAs (Sec. 6).

## 2 Background

In this section, we formalize the problem setup of generative MIAs and introduce the necessary geometric concepts and notations. A detailed discussion of related work is provided in Appendix A.

**Problem Setup of Generative MIAs.** Let the ambient space be $\mathcal{X} = \mathbb{R}^d$, and the private label space be $\mathcal{Y}_{\text{pri}} = \{1, \ldots, C\}$. The target model $f(\cdot; \boldsymbol{\theta}) : \mathcal{X} \to \mathbb{R}^C$ is a classifier that outputs *class logits*, trained on a private dataset $\mathcal{D}_{\text{pri}}$ sampled from distribution $p_{\text{pri}}(\mathbf{x}, y)$. We presume the *manifold hypothesis*: the private data distribution $p_{\text{pri}}$ is supported on a low-dimensional submanifold $\mathcal{M}_{\text{pri}} \subset \mathbb{R}^d$ with intrinsic dimension $k \ll d$. In MIAs, the adversary aims to synthesize inputs that reveal class-sensitive features of the private training data for a target class $y \in \mathcal{Y}_{\text{pri}}$. The adversary is assumed to have white-box access to the model $f$, as well as general knowledge of the data domain (*e.g.,* the data consists of facial images), but no direct access to the private dataset $\mathcal{D}_{\text{pri}}$.

MIAs are typically framed as an optimization problem: given a target class $y$, the adversary seeks for an input $\mathbf{x} \in \mathcal{X}$ that maximizes the likelihood of $y$ under model $f$ [Fredrikson et al., 2015]. However, when $f$ is a DNN trained on high-dimensional data, direct optimization in the ambient space $\mathcal{X}$ often results in unrealistic samples that lack semantic relevance [Szegedy et al., 2014], due to the ill-posed nature of the problem. To address this challenge, Zhang et al. [2020] proposed a two-stage generative model inversion framework, which we outline below.

In the first stage, the adversary collects a public auxiliary dataset $\mathcal{D}_{\text{aux}}$ drawn from a distribution $p_{\text{aux}}$, with a label set disjoint from that of the private training dataset (*i.e.,* $\mathcal{Y}_{\text{aux}} \cap \mathcal{Y}_{\text{pri}} = \emptyset$). The distribution $p_{\text{aux}}$ is assumed to be supported on a submanifold $\mathcal{M}_{\text{aux}}$ that approximates the private data manifold $\mathcal{M}_{\text{pri}}$, even though $p_{\text{pri}}$ and $p_{\text{aux}}$ may differ. A GAN is then trained on $\mathcal{D}_{\text{aux}}$ to estimate $p_{\text{aux}}$, consisting of a generator $G : \mathcal{Z} \to \mathcal{X}$ that maps latent variables $\mathbf{z} \in \mathcal{Z} = \mathbb{R}^k$ to samples $\mathbf{x} \in \mathcal{M}_{\text{aux}}$, and a discriminator $D : \mathcal{X} \to \mathbb{R}$ that distinguishes real from generated data. In the second stage, the adversary performs attack optimization in the latent space $\mathcal{Z}$ of the generator $G$, effectively restricting the search space to the manifold $\mathcal{M}_{\text{aux}}$, which can be formulated as:

$$\mathbf{z}^* = \arg \min_{\mathbf{z}} \mathcal{L}_{\text{inv}}(\mathbf{z}; y, f, G, D) = \mathcal{L}_{\text{cls}}(\mathbf{z}; y, f, G) + \lambda \mathcal{L}_{\text{prior}}(\mathbf{z}; G, D). \tag{1}$$

Here, $\mathcal{L}_{\text{cls}}$ denotes the *inversion-time* classification loss, *e.g.,* the logit loss $-f_y(G(\mathbf{z}))$ [Nguyen et al., 2023b], which drives the optimization toward a synthetic sample $\mathbf{x}^* = G(\mathbf{z}^*)$ that maximally activates class $y$. The term $\mathcal{L}_{\text{prior}}$ regularizes the latent code $\mathbf{z}$, promoting plausible generations. The hyperparameter $\lambda$ controls the trade-off between the two loss terms.

**Geometric Preliminaries and Notation.** Let $\mathcal{M} \subset \mathbb{R}^d$ be a $k$-dimensional differentiable manifold. At any point $\mathbf{x} \in \mathcal{M}$, the tangent space $T_{\mathbf{x}}\mathcal{M}$ is a $k$-dimensional linear subspace of $\mathbb{R}^d$ that locally approximates $\mathcal{M}$, capturing the directions of infinitesimal motion that remain on the manifold. To formalize projections onto this space, we denote by $\mathbf{P}_{\mathbf{x}} \in \mathbb{R}^{d \times d}$ the projection matrix that projects vectors in $\mathbb{R}^d$ onto $T_{\mathbf{x}}\mathcal{M}$. When $\mathbf{P}_{\mathbf{x}}$ is symmetric and idempotent, it defines an orthogonal projection.

Now consider a differentiable generator $G : \mathcal{Z} \to \mathcal{X}$. For any latent vector $\mathbf{z} \in \mathcal{Z}$, the Jacobian matrix $J_G(\mathbf{z}) = \frac{\partial G}{\partial \mathbf{z}} \in \mathbb{R}^{d \times k}$ characterizes how infinitesimal perturbations in the latent space map to

changes in data space. If $J_G(\mathbf{z})$ has full column rank $k$, then the image of G forms a $k$-dimensional differentiable manifold $\mathcal{M} \subset \mathbb{R}^d$ [Lee, 2003]. Moreover, the column span (*i.e.,* range) of $J_G(\mathbf{z})$ equals the tangent space at the corresponding point $\mathbf{x} = G(\mathbf{z})$:

$$T_{\mathbf{x}}\mathcal{M} = \text{span}\left\{\frac{\partial G}{\partial \mathbf{z}_1}, \ldots, \frac{\partial G}{\partial \mathbf{z}_k}\right\} =: \text{Range}(J_G(\mathbf{z})). \tag{2}$$

The Jacobian $J_G$ plays a key role in our analysis of how loss gradients interact with the generator manifold in the context of generative model inversion, as detailed in the next section.

## 3 A Geometric Lens for Understanding Generative MIAs

In this section, we analyze how generative model inversion exploits private information encoded in a target model $f$ to reconstruct input samples. Central to this process are the the inversion-time loss gradients $\nabla_{\mathbf{x}}\mathcal{L}_{\text{cls}}(f(\mathbf{x}), y)$ (with $\mathbf{x} = G(\mathbf{z})$), which are backpropagated to optimize the latent variable $\mathbf{z}$ (see Fig. 1(a)). Empirically, we find that these gradients are often highly noisy (see Figs. 1(b) and 2), with many components misaligned with the intrinsic structure of the generator manifold. This observation may explain why direct optimization in the ambient space frequently leads to semantically meaningless samples [Fredrikson et al., 2015, Zhang et al., 2020, Wang et al., 2021a]. To better understand how the generator G processes this gradient signal, we analyze the transformation of the gradient via the Jacobian $J_G$. By the chain rule, the loss gradients *w.r.t.* the latent variable $\mathbf{z}$ can be expressed as:

$$\nabla_{\mathbf{z}}\mathcal{L}_{\text{cls}} = (J_G)^{\top}\nabla_{\mathbf{x}}\mathcal{L}_{\text{cls}} \in \mathbb{R}^k$$
$$= \left[\left\langle\frac{\partial G}{\partial \mathbf{z}_1}, \nabla_{\mathbf{x}}\mathcal{L}_{\text{cls}}\right\rangle, \ldots, \left\langle\frac{\partial G}{\partial \mathbf{z}_k}, \nabla_{\mathbf{x}}\mathcal{L}_{\text{cls}}\right\rangle\right]^{\top}.$$

Thus, $\nabla_{\mathbf{z}}\mathcal{L}_{\text{cls}}$ can be interpreted as expressing the ambient loss gradients $\nabla_{\mathbf{x}}\mathcal{L}_{\text{cls}}$ in the basis formed by the columns of the generator's Jacobian (*i.e.,* the tangent basis at $\mathbf{x}$). In other words, each component of the latent gradient represents the directional derivative of the loss along one of the generator's valid, manifold-constrained directions. This "pullback" maps the high-dimensional

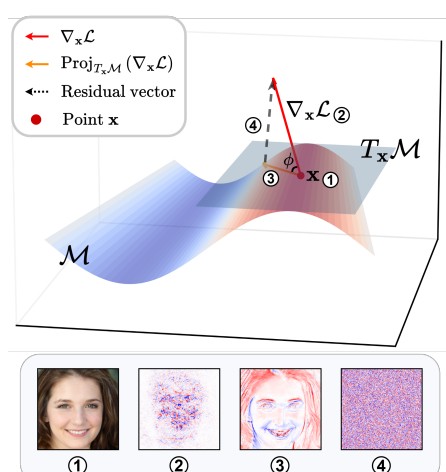

Figure 2: **Geometric interpretation of loss gradients projection onto the generator manifold.** The generative model inversion process implicitly denoises the loss gradients $\nabla_{\mathbf{x}}\mathcal{L}$ by projecting them onto the tangent space $T_{\mathbf{x}}\mathcal{M}$ of the generator manifold $\mathcal{M}$. The bottom panel illustrates the reconstructed image, its inversion-time loss gradients, the manifold-projected gradients, and the residual component.

loss gradient into the latent space, yielding a structured signal aligned with the generator manifold. To understand how these latent gradients influence updates in data space, we now analyze their pushforward by applying a first-order Taylor approximation:

$$G(\mathbf{z} - \eta\nabla_{\mathbf{z}}\mathcal{L}_{\text{cls}}) - G(\mathbf{z}) \approx -\eta J_G \nabla_{\mathbf{z}}\mathcal{L}_{\text{cls}}.$$

Here, $\eta$ denotes the step size for updating $\mathbf{z}$. The term $J_G\nabla_{\mathbf{z}}\mathcal{L}_{\text{cls}}$ represents a linear combination of the tangent basis vectors at $\mathbf{x}$, where $\nabla_{\mathbf{z}}\mathcal{L}_{\text{cls}}$ serves as the coordinate vector in this basis. As a result, it lies entirely within the tangent space $T_{\mathbf{x}}\mathcal{M}^2$ (see Eq. (2)). More importantly, the resulting vector can be interpreted as the projection of the ambient gradients $\nabla_{\mathbf{x}}\mathcal{L}_{\text{cls}}$ onto the tangent space $T_{\mathbf{x}}\mathcal{M}$:

$$\text{Proj}_{T_{\mathbf{x}}\mathcal{M}}(\nabla_{\mathbf{x}}\mathcal{L}_{\text{cls}}) = J_G\nabla_{\mathbf{z}}\mathcal{L}_{\text{cls}} = \left[J_G(J_G)^{\top}\right]\nabla_{\mathbf{x}}\mathcal{L}_{\text{cls}} = \widetilde{\mathbf{P}}_{\mathbf{x}}\nabla_{\mathbf{x}}\mathcal{L}_{\text{cls}},$$

where $\widetilde{\mathbf{P}}_{\mathbf{x}} = J_G(J_G)^{\top}$ is an unnormalized projector onto the tangent space $T_{\mathbf{x}}\mathcal{M}$. This projection operation has a critical denoising effect: it preserves only the gradient components aligned with the tangent space of the generator manifold, while filtering out directions that deviate from it (see Fig. 2). Thus, backpropagation through the generator fundamentally acts as a geometric filter, allowing optimization to proceed along semantically meaningful directions.

---

[2]For notational simplicity, we use $\mathcal{M}$ to denote $\mathcal{M}_{\text{aux}}$ throughout this section.

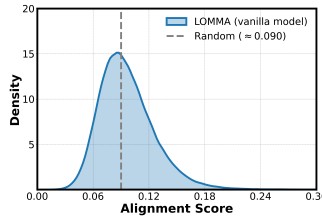
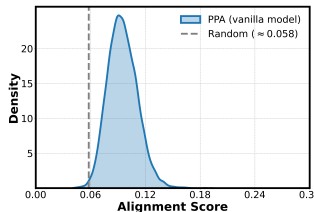
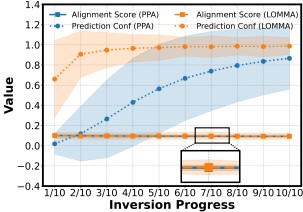

(a) Low resolution (DCGAN)  (b) High resolution (StyleGAN)  (c) Inversion-phase dynamics

Figure 3: **gradient-manifold alignment during the inversion process.** **(a)** Alignment score distribution in the low-resolution setting using LOMMA with a DCGAN trained on CelebA. **(b)** High-resolution counterpart using PPA with a StyleGAN trained on FFHQ. In both cases, alignment scores are only slightly above those of random vectors, suggesting weak alignment along the generator manifold. **(c)** Evolution of the average alignment score and prediction confidence over the inversion process. While the model's prediction confidence steadily increases, the gradient-manifold alignment remains consistently low, indicating no direct dependence between the two. For additional details on this experiment, as well as results from other attack methods, refer to Appendix D.6.

To assess how well the loss gradient $\nabla_{\mathbf{x}}\mathcal{L}_{\text{cls}}$ aligns with informative directions, we measure its alignment with the tangent space $T_{\mathbf{x}}\mathcal{M}$ at point $\mathbf{x}$. Specifically, we quantify this alignment by computing the *cosine of the angle* between the loss gradients and the projection onto $T_{\mathbf{x}}\mathcal{M}$. Note that while $\widetilde{\mathbf{P}}_{\mathbf{x}}$ performs a projection onto $T_{\mathbf{x}}\mathcal{M}$, it is not a valid orthogonal projection operator unless the columns of $J_{\mathrm{G}}$ are orthonormal. To construct an orthogonal projector, we perform singular value decomposition (SVD) on the $J_{\mathrm{G}}$: $J_{\mathrm{G}} = \mathbf{U}\boldsymbol{\Sigma}\mathbf{V}^{\top}$, where $\mathbf{U} \in \mathbb{R}^{d \times d}$, $\boldsymbol{\Sigma} \in \mathbb{R}^{d \times k}$ and $\mathbf{V} \in \mathbb{R}^{k \times k}$. Let $\mathbf{U}_k \in \mathbb{R}^{d \times k}$ denote the matrix consisting of the first $k$ left-singular vectors, which form an orthonormal basis for $\text{Range}(J_{\mathrm{G}})$, *i.e.,* the tangent space $T_{\mathbf{x}}\mathcal{M}$ (see Eq. (2)). The corresponding orthogonal projection matrix is then given by $\mathbf{P}_{\mathbf{x}} = \mathbf{U}_k\mathbf{U}_k^{\top}$. Consequently, we compute the cosine of the angle $\phi$ between the loss gradients and the projection on the tangent space as:

$$\cos(\phi) = \left\|\mathbf{P}_{\mathbf{x}}\,\nabla_{\mathbf{x}}\mathcal{L}_{\text{cls}}\right\| / \left\|\nabla_{\mathbf{x}}\mathcal{L}_{\text{cls}}\right\|. \tag{3}$$

We refer to this quantity as the *alignment score*, denoted $\text{AS}(\nabla_{\mathbf{x}}\mathcal{L}_{\text{cls}}) := \cos(\phi)$, which quantifies the extent to which the loss gradient lies within the tangent space at point $\mathbf{x}$. Higher values correspond to smaller angles and thus indicate stronger alignment. When evaluating Eq. (3), it is important to note that even random vectors exhibit non-zero projections onto the tangent space purely due to geometric effects. In expectation, a random vector aligns with a $k$-dimensional subspace with a magnitude of approximately $\sqrt{k/d}$ [Vershynin, 2018]. To assess the informativeness of the loss gradients, we track the alignment score throughout the inversion process. Empirically, we observe that in models trained with standard supervision, the alignment score remains consistently low (see Fig. 3). This suggests that the loss gradients frequently point in directions misaligned with the underlying data manifold, and therefore carry limited semantically meaningful information for guiding inversion.

## 4 Does Gradient-Manifold Alignment Indicate MIA Vulnerability?

Motivated by the previous empirical findings and the intuition that stronger alignment with the generator manifold reflects more informative gradients, we propose the following hypothesis:

*Models tend to be more vulnerable to MIAs when their loss gradients are more aligned with the tangent space of the generator manifold.*

To validate our hypothesis, we aim to design a training objective that promotes stronger alignment between loss gradients and the generator manifold during inversion. A key challenge, however, is that this alignment is *not directly accessible* during training. To bridge this gap, we analyze the inversion-time classification loss, which is the only term that directly interacts with the target model $f$ to extract private information. By the chain rule, its gradients can be expressed as a linear combination of input gradients (*i.e.,* the gradients of model outputs *w.r.t.* inputs) [Srinivas and Fleuret, 2021, Bhalla et al., 2023]. Formally, let $f(\mathbf{x}) = [f_1(\mathbf{x}), f_2(\mathbf{x}), \ldots, f_C(\mathbf{x})]$ denote the model's logits for $C$ classes. The inversion-time classification loss $\mathcal{L}_{\text{cls}}(f(\mathbf{x}), y)$ is a function of these logits, *i.e.,*

$\mathcal{L}_{\text{cls}}(f(\mathbf{x}), y) = \mathcal{L}_{\text{cls}}\big(f_1(\mathbf{x}), f_2(\mathbf{x}), \ldots, f_C(\mathbf{x})\big)$. Thus, by the chain rule, we obtain:

$$\nabla_{\mathbf{x}} \mathcal{L}_{\text{cls}} = \sum_{i=1}^{C} \frac{\partial \mathcal{L}_{\text{cls}}}{\partial f_i} \nabla_{\mathbf{x}} f_i. \tag{4}$$

In other words, the loss gradient is a weighted sum of input gradients, where the weights $\frac{\partial \mathcal{L}_{\text{cls}}}{\partial f_i}$ quantify the sensitivity of the loss to each logit. This structural insight allows us to shift our focus from loss gradients to input gradients, which are directly accessible during training [Sundararajan et al., 2017, Dwivedi et al., 2023]. Moreover, if the input gradients align well with the data manifold, then by construction, the loss gradients will also exhibit improved alignment. Thus, during training, we propose to encourage alignment between input gradients and the tangent space of the data manifold, thereby indirectly promoting alignment of loss gradients during the inversion phase, making alignment-aware training feasible without requiring access to the inversion process.

**Gradient-Manifold Alignment Training.** Building on the above analysis, we propose a novel training objective to validate our hypothesis. It consists of two components: (1) the standard cross-entropy (CE) loss for training-time classification, and (2) an auxiliary term that explicitly encourages the model's input gradients to align with the estimated tangent space of the data manifold. Crucially, the second term leverages the fact that the input gradients of the classifier, $\nabla_{\mathbf{x}} f_i(\mathbf{x}; \boldsymbol{\theta})$, are differentiable *w.r.t.* model parameters $\boldsymbol{\theta}$, and can therefore be directly optimized during training.

To estimate the tangent space of the data manifold, we leverage a powerful pre-trained variational autoencoder (VAE), specifically, the one used in Stable Diffusion [Rombach et al., 2022]. Trained on large-scale datasets [Kuznetsova et al., 2020, Schuhmann et al., 2022], this VAE provides a strong approximation of the natural image manifold. A VAE consists of an encoder $\mathcal{E}$ and a decoder $\mathcal{D}$. Given an input image $\mathbf{x}$, the encoder maps it to a latent vector $\mathbf{z} = \mathcal{E}(\mathbf{x})$, and the decoder reconstructs the image as $\hat{\mathbf{x}} = \mathcal{D}(\mathbf{z})$[3]. The decoder $\mathcal{D}$ implicitly defines a data manifold with intrinsic dimension equal to that of the latent space. Then, for any data point $\mathbf{x} \in \mathcal{D}_{\text{pri}}$, we estimate its tangent space via the Jacobian of $\mathcal{D}$, approximated by $\text{Range}(J_{\mathcal{D}}(\mathbf{z}))$. Following the method in Sec. 3, we construct the orthogonal projection matrix $\mathbf{P}_{\mathbf{x}}$ onto the tangent space at $\mathbf{x}$. To promote alignment between the model's input gradients and the data manifold, we propose the following training objective:

$$\mathcal{L}_{\text{align}}(\boldsymbol{\theta}) = \mathbb{E}_{(\mathbf{x}, y) \sim \mathcal{D}_{\text{pri}}} \left[ \mathcal{L}_{\text{CE}}(f(\mathbf{x}; \boldsymbol{\theta}), y) - \beta \frac{1}{C} \sum_{i=1}^{C} \frac{\left\| \mathbf{P}_{\mathbf{x}} \nabla_{\mathbf{x}} f_i(\mathbf{x}; \boldsymbol{\theta}) \right\|}{\left\| \nabla_{\mathbf{x}} f_i(\mathbf{x}; \boldsymbol{\theta}) \right\|} \right], \tag{5}$$

where the first term is the standard cross-entropy loss for classification, and the second term encourages the input gradients to align with the estimated tangent space. The hyperparameter $\beta$ controls the trade-off between the two objectives. However, computing this alignment term requires $C$ projection operations per training example (one per class logit), which can become computationally expensive when the input dimension is high or the number of classes is large. To reduce this cost, we derive the following upper bound for the alignment promotion term (see Appendix B for proof):

$$-\frac{\left\| \mathbf{P}_{\mathbf{x}} \sum_{i=1}^{C} \nabla_{\mathbf{x}} f_i(\mathbf{x}; \boldsymbol{\theta}) \right\|}{\left\| \sum_{i=1}^{C} \nabla_{\mathbf{x}} f_i(\mathbf{x}; \boldsymbol{\theta}) \right\|} \geq -\frac{1}{C} \sum_{i=1}^{C} \frac{\left\| \mathbf{P}_{\mathbf{x}} \nabla_{\mathbf{x}} f_i(\mathbf{x}; \boldsymbol{\theta}) \right\|}{\left\| \nabla_{\mathbf{x}} f_i(\mathbf{x}; \boldsymbol{\theta}) \right\|}. \tag{6}$$

This inequality allows us to define a more efficient surrogate objective that only requires a single projection operation per data point (the algorithmic implementation is provided in Appendix C):

$$\mathcal{L}_{\text{align}}(\boldsymbol{\theta}) = \mathbb{E}_{(\mathbf{x}, y) \sim \mathcal{D}_{\text{pri}}} \left[ \mathcal{L}_{\text{CE}}\big(f(\mathbf{x}; \boldsymbol{\theta}), y\big) - \beta \frac{\left\| \mathbf{P}_{\mathbf{x}} \sum_{i=1}^{C} \nabla_{\mathbf{x}} f_i(\mathbf{x}; \boldsymbol{\theta}) \right\|}{\left\| \sum_{i=1}^{C} \nabla_{\mathbf{x}} f_i(\mathbf{x}; \boldsymbol{\theta}) \right\|} \right]. \tag{7}$$

Empirical results confirm that this alignment-aware training increases the model's vulnerability to generative MIAs, thereby validating our hypothesis (see Sec. 6.2). Moreover, this finding motivates the design of a training-free method to further enhance gradient-manifold alignment during the inversion process, as detailed in the next section.

---

[3]To align with VAE literature, we use $\mathcal{D}$ to denote the VAE decoder, despite its earlier use for the dataset in Sec. 2. The latent representation $\mathbf{z}$ is similarly reused for notational convenience.

## 5 Enhancing Gradient-Manifold Alignment Without Training

Motivated by the previous observations, we propose AlignMI, a *training-free* approach to enhance gradient-manifold alignment during the inversion process. The core idea is geometric: since informative gradients lie along the tangent space of the generator manifold, we aim to suppress off-manifold components and enhance alignment with semantically meaningful directions. To this end, rather than relying on a single gradient estimate at a synthetic input $\mathbf{x}$, we sample multiple variants of $\mathbf{x}$ within a local neighborhood and average their corresponding loss gradients. This averaging process attenuates noisy, off-manifold directions while amplifying consistent components aligned with the manifold, yielding a more semantically meaningful and geometrically coherent signal. Formally, let $N(\mathbf{x}) \subset \mathbb{R}^d$ denote a measurable neighborhood around $\mathbf{x}$, and let $p(\cdot \mid \mathbf{x})$ be a probability distribution supported on $N(\mathbf{x})$. We define the smoothed, alignment-enhanced gradient as:

$$\widetilde{\nabla}\mathcal{L}(\mathbf{x}) = \mathbb{E}_{\mathbf{x}' \sim p(\cdot \mid \mathbf{x})}\big[\nabla\mathcal{L}(\mathbf{x}')\big]. \tag{8}$$

This technique is entirely training-free and can be applied directly at the inversion time. We instantiate AlignMI with two concrete strategies for sampling from the neighborhood $N(\mathbf{x})$.

**(1) Perturbation-Averaged Alignment (PAA).** In this realization, the neighborhood distribution is defined as an isotropic Gaussian centered at the synthetic input:

$$p(\cdot \mid \mathbf{x}) = \mathcal{N}(\mathbf{x}, \sigma^2 \mathbf{I}).$$

This corresponds to sampling within a spherical region around $\mathbf{x}$, smoothing the gradient by averaging over random perturbations. The process suppresses high-frequency, noisy components that are likely to deviate from the generator manifold.

**(2) Transformation-Averaged Alignment (TAA).** Alternatively, in this realization, we define the distribution as uniform over a set of semantically invariant transformations:

$$p(\cdot \mid \mathbf{x}) = \text{Uniform}\{\tau(\mathbf{x}) \mid \tau \in \mathcal{T}\},$$

where $\mathcal{T}$ is a predefined set of semantic-preserving transformations, such as random cropping, flipping, or affine warping. This formulation captures local perturbations along the manifold, encouraging alignment with directions that preserve perceptual consistency and geometric semantics. Both PAA and TAA are model-agnostic and fully post hoc. Their algorithmic implementations are provided in Appendix C. As demonstrated in Sec. 6.3, incorporating either strategy consistently improves inversion performance by producing loss gradients that are better aligned with the generator manifold.

## 6 Experiments

In this section, we first validate the hypothesis proposed in Sec. 4, followed by a comprehensive evaluation of the training-free AlignMI approach introduced in Sec. 5. Our experiments focus on real-world face recognition tasks. To ensure computational efficiency, we perform hypothesis validation in the low-resolution setting ($64 \times 64$), where tangent space estimation is tractable. For the method evaluation, we compare the performance of state-of-the-art generative MIAs before and after integrating our proposed techniques, *i.e.,* PAA and TAA. Specifically, in the high-resolution setting ($224 \times 224$), we evaluate on PPA [Struppek et al., 2022]. For the low-resolution setting, we consider GMI (LOMMA) with StyleGAN [Zhang et al., 2020, Karras et al., 2020, Nguyen et al., 2023b], KEDMI (LOMMA) with DCGAN [Chen et al., 2021], and PLG-MI [Yuan et al., 2023]. In addition, we evaluate the performance of these methods against strong MIA defenses, including BiDO [Peng et al., 2022], NegLS [Struppek et al., 2024], and TL-DMI [Ho et al., 2024].

### 6.1 Experimental Setup

We begin with a brief overview of the experimental setup; refer to Appendix D for details.

**Datasets and Models.** In line with existing MIA literature, we use the CelebA [Liu et al., 2015], FaceScrub [Ng and Winkler, 2014], and FFHQ datasets [Karras et al., 2019]. These datasets are divided into two parts: the private training dataset $\mathcal{D}_{\text{pri}}$ and the public auxiliary dataset $\mathcal{D}_{\text{aux}}$, with no overlapping classes. For high-resolution tasks, we use ResNet-18 [He et al., 2016], DenseNet-121 [Huang et al., 2017] and ResNeSt-50 [Zhang et al., 2022] as target models. For low-resolution

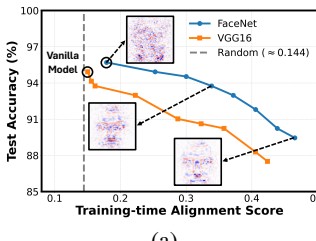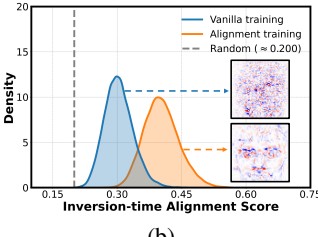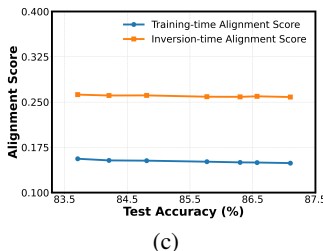

(a)  (b)  (c)

Figure 4: **Empirical evaluation of gradient-manifold alignment. (a)** Test accuracy vs. training-time alignment score ($AS_{tr}$) for models sampled during alignment-aware training. Insets show input gradient visualizations for models with varying degrees of alignment. **(b)** Distribution of inversion-time alignment scores ($AS_{inv}$) for the vanilla model compared to the alignment-aware model. **(c)** Average alignment scores $AS_{tr}$ and $AS_{inv}$ across models with varying test accuracy. Enlarged versions of (a) and (b), along with experimental details, are provided in Appendix D.7.

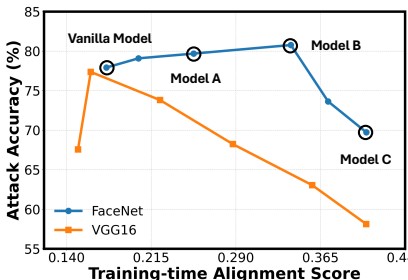

Figure 5: **MIA success on vanilla and alignment-aware models with different** $AS_{tr}$.

Table 1: **Alignment score, predictive accuracy, and inversion vulnerability for vanilla and three alignment-aware models.**

| Training Variant | $AS_{tr}$ | Test Acc | Acc@1 | KNN Dist |
|---|---|---|---|---|
| Vanilla | 0.175 | 96.53 | 77.92 | 1452.20 |
| Model A | 0.253 | 94.92 | 79.68 | 1413.53 |
| Model B | 0.339 | 93.75 | 80.76 | 1408.00 |
| Model C | 0.406 | 91.80 | 69.72 | 1613.96 |

tasks, we use VGG16 [Simonyan and Zisserman, 2015] and FaceNet [Wang et al., 2021b] as target models. Training details for these models are provided in Appendix D.2. A summary of the attack methods, target models, and datasets used is provided in Tab. 3.

**Evaluation Metrics.** (1) To quantify the alignment between gradients and the image manifold, we report two metrics: training-time alignment scores based on input gradients ($AS_{tr}$), and inversion-time alignment scores based on loss gradients ($AS_{inv}$), as defined in Eq. (3). (2) To evaluate the inversion performance of MIAs, we follow standard metrics in the literature [Zhang et al., 2020], including top-1 (Acc@1) and top-5 (Acc@5) attack accuracy, as well as K-Nearest Neighbors Distance (KNN Dist). Details for these metrics are provided in Appendix D.5.

### 6.2 Empirical Validation of the Hypothesis

In this subsection, we empirically validate the hypothesis introduced in Sec. 4, by comparing standard (vanilla) models with alignment-aware models trained using the objective defined in Eq. (7). For this evaluation, we adopt GMI (LOM) as the inversion method, using a StyleGAN as the prior.

**Gradient-manifold Alignment Analysis.** Specifically, we fine-tune the pre-trained vanilla VGG16 and FaceNet models using the alignment-aware training objective, resulting in multiple models with varying training-time alignment scores ($AS_{tr}$). As shown in Fig. 4(a), the vanilla models exhibit low alignment scores (approximately 0.15 and 0.18), which marginally exceed those expected from random vectors, indicating weak alignment between input gradients and the data manifold. As fine-tuning progresses, $AS_{tr}$ steadily increases, and gradient visualizations reveal a corresponding rise in semantically meaningful features. Notably, this increase in alignment is accompanied by a gradual decline in test accuracy, suggesting an empirical trade-off between gradient-manifold alignment and predictive performance. This trend is consistent across both architectures. We hypothesize that this trade-off could stem from the inherent limitations of modern deep neural network architectures or the implicit biases introduced by stochastic gradient-based optimization.

Fig. 4(b) compares the distribution of inversion-time alignment scores ($AS_{inv}$) between the FaceNet vanilla model and the alignment-aware model (corresponding to Model B in Fig. 5). The Model

trained with the alignment-aware objective exhibits significantly higher $AS_{inv}$ values, demonstrating that promoting gradient-manifold alignment during training leads to stronger alignment at inversion time. This validates the effectiveness of our training strategy. Additionally, the inversion loss gradient visualizations also reveal clearer and more semantically meaningful structures.

**Model Inversion Vulnerability Analysis.** We further evaluate the vulnerability of both vanilla and alignment-aware models to the GMI (LOM) attack method. As shown in Fig. 5, model inversion vulnerability initially increases with training-time alignment score ($AS_{tr}$), reaching a peak before declining. To understand this behavior, we first recall that previous work [Zhang et al., 2020] has shown that models with higher predictive power (*i.e.,* test accuracy) tend to be more susceptible to generative MIAs, whereas models with lower test accuracy are generally more resistant. If gradient-manifold alignment were unrelated to model inversion vulnerability, one would expect a monotonic decline in attack accuracy with increasing $AS_{tr}$, due to the corresponding drop in test accuracy (Fig. 4(a)). However, Fig. 5 instead shows a non-monotonic trend.

This trend arises because improvements in gradient-manifold alignment create a new attack surface, leading to increased model inversion vulnerability. At early stages, the benefits of improved alignment outweigh the negative impact of reduced test accuracy, hence attack accuracy rises. Beyond a certain point, however, the adverse effects of declining test accuracy become dominant, and attack accuracy begins to decline. This trend holds consistently across both architectures we studied. These findings suggest that, for models with comparable test accuracy, those with better gradient-manifold alignment tend to be more vulnerable to MIAs, thus supporting our main hypothesis.

To illustrate this trend in detail, we examine three representative alignment-aware models and report their training-time alignment scores, test accuracies, and MIA vulnerabilities. Results in Tab. 1 show that models A and B, despite having lower test accuracy than the vanilla baseline, exhibit greater vulnerability to MIAs. This increased susceptibility is attributable to their higher $AS_{tr}$, which produces more informative loss gradients during inversion. In contrast, model C, with a higher alignment score but lower test accuracy, shows reduced vulnerability, suggesting that excessive alignment may come at the cost of generalization and thereby diminish the attack surface.

Moreover, we study the correlation between model predictive power and the gradient-manifold alignment. As shown in Fig. 4(c), both $AS_{tr}$ and $AS_{inv}$ remain relatively steady across models with varying predictive performance, indicating that model predictive power and gradient-manifold alignment exhibit little correlation. These results suggest that gradient-manifold alignment captures a complementary aspect of model inversion vulnerability—one not explained by predictive power alone—and offer new insights into the factors underlying privacy risks in machine learning models.

## 6.3 Evaluation of Proposed Methods

In this subsection, we evaluate the effectiveness of our training-free AlignMI approach by comparing the inversion performance of the PPA method before and after integration with its two realizations, PAA and TAA. This evaluation focuses on the high-resolution setting, representing a more realistic and challenging attack scenario. Additional experiments, including results on low-resolution MIAs, evaluations under SOTA MIA defenses, and ablation studies, are provided in Appendix E.

For all experiments, we configure PAA with Gaussian perturbations of standard deviation $\sigma$ set to $5\%$ of the synthesized images' dynamic range. For TAA, we apply standard semantic-preserving transformations, including random resized cropping with scale $[0.8, 1.0]$ and aspect ratio $[0.9, 1.1]$), horizontal flipping with a probability of 0.5, and random rotations within $\pm 5°$. For both methods, we average the loss gradients over 50 samples to approximate the expectation in Eq. (8).

We conduct three independent runs for both the baseline and our method. The mean results are reported in Tab. 2, and the complete results are provided in Tab. 4 in Appendix E. The results demonstrate that our methods consistently enhance inversion performance across all setups, yielding higher attack accuracy and lower KNN distance, thus validating their effectiveness. Notably, TAA outperforms PAA in most cases. This is because PAA improves alignment by adding noise perturbations to loss gradients, which can reduce prediction confidence, as models are typically not trained on noisy inputs. In contrast, TAA uses semantic-preserving augmentations, which maintain input realism and avoid this trade-off. Visualizations of gradient images and reconstructed samples for the target models are provided in Appendix E.4 and Appendix E.5, respectively.

Table 2: Comparison of inversion performance with PPA in the high-resolution setting. $\mathcal{D}_{pri}$ = CelebA or FaceScrub, GANs are pre-trained on $\mathcal{D}_{aux}$ = FFHQ. The symbol ↓ (or ↑) indicates that smaller (or larger) values are preferred, and the green numbers represent the performance improvement. The results are averaged over three independent runs.

| Target Model | Method | CelebA | | | | FaceScrub | | | |
| | | Acc@1↑ | Acc@5↑ | KNN Dist↓ | Ratio↓ | Acc@1↑ | Acc@5↑ | KNN Dist↓ | Ratio↓ |
|---|---|---|---|---|---|---|---|---|---|
| **ResNet-18** | PPA | 85.63 | 95.12 | 0.693 | / | 81.57 | 94.85 | 0.796 | / |
| | + PAA (ours) | 88.75 (+3.12) | 96.59 | 0.669 (-0.024) | 1.50 | 83.97 (+2.40) | 95.78 | 0.777 (-0.019) | 1.55 |
| | + TAA (ours) | 91.68 (+6.05) | 97.68 | 0.662 (-0.031) | 1.61 | 93.68 (+12.11) | 98.84 | 0.691 (-0.105) | 1.61 |
| **DenseNet-121** | PPA | 82.22 | 93.26 | 0.708 | / | 75.66 | 90.91 | 0.786 | / |
| | + PAA (ours) | 85.64 (+3.42) | 95.16 | 0.684 (-0.024) | 2.82 | 80.70 (+5.04) | 93.40 | 0.761 (-0.025) | 2.82 |
| | + TAA (ours) | 87.88 (+5.66) | 96.20 | 0.687 (-0.021) | 2.87 | 86.54 (+10.88) | 95.12 | 0.712 (-0.074) | 2.93 |
| **ResNeSt-50** | PPA | 70.75 | 87.43 | 0.793 | / | 71.58 | 90.60 | 0.827 | / |
| | + PAA (ours) | 75.71 (+4.96) | 90.48 | 0.764 (-0.029) | 2.93 | 73.38 (+1.80) | 91.34 | 0.807 (-0.020) | 3.12 |
| | + TAA (ours) | 79.19 (+8.44) | 92.28 | 0.761 (-0.032) | 3.12 | 84.38 (+12.80) | 96.04 | 0.753 (-0.074) | 3.13 |

## 7 Discussion

**Limitations.** Our experiments validate the proposed hypothesis in the low-resolution setting, where gradient-manifold alignment-aware training is currently feasible only at this scale. We observe an empirical trade-off between alignment and predictive performance, suggesting that stronger alignment may come at the cost of impairing model generalization. However, due to computational limitations, we are unable to examine whether this trend persists in high-resolution settings. In particular, high-resolution inputs of size $224 \times 224 \times 3$ produce latent representations of size $28 \times 28 \times 4$ from the VAE encoder, resulting in a decoder Jacobian of size $150{,}528 \times 3136$. This is roughly 150 times larger than in the low-resolution case, rendering tangent space estimation computationally and memory intensive. Moreover, the underlying cause of the observed alignment-accuracy trade-off remains unclear and warrants more systematic in future work.

**Broader Impacts.** From a geometric perspective, our analysis identifies a previously underexplored factor contributing to model inversion vulnerability, complementing existing explanations focused on predictive power. This perspective offers new insight into the mechanisms that give rise to privacy risks in machine learning models. Beyond its technical implications, the AlignMI approach also raises important ethical considerations: if misused, it could increase the likelihood of exposing sensitive training data. Conversely, this geometric understanding provides a foundation for principled defenses against generative MIAs. In particular, reducing gradient-manifold alignment may serve as an effective strategy for mitigating such vulnerabilities.

## 8 Conclusion

In this work, we investigate the underlying mechanism of generative model inversion from a geometric perspective. We show that generative MIAs implicitly denoise loss gradients by projecting them onto the tangent space of the generator manifold, preserving informative on-manifold directions while filtering out noisy off-manifold components. Building on this insight, we identified a previously underexplored vulnerability: models with loss gradients align more strongly with the generator manifold tend to be more susceptible to inversion attacks. We validated this hypothesis using a novel training objective that explicitly promotes gradient-manifold alignment. Finally, we propose AlignMI, a training-free approach to enhance such alignment during inversion, and demonstrate its effectiveness through extensive experiments across multiple attack methods.

## Acknowledgments

XP, FFY, and BH were supported by NSFC General Program No. 62376235, RGC Young Collaborative Research Grant No. C2005-24Y, RGC General Research Fund No. 12200725, Guangdong Basic and Applied Basic Research Foundation Nos. 2022A1515011652 and 2024A151501239, and HKBU CSD Departmental Incentive Scheme. TLL is partially supported by the following Australian Research Council projects: FT220100318, DP220102121, LP220100527, LP220200949.

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

# Appendix

# A  Related Work

*Model inversion attacks* (MIAs) were first introduced by Fredrikson et al. [2014], who demonstrated the reconstruction of private data in simple regression tasks using shallow models. Their pioneering attack algorithm aimed to infer sensitive attributes, such as genetic markers, via input space optimization, assuming access to both the linear target model and auxiliary information. This work highlighted the privacy risks inherent in exposing model predictions. Building on this, Fredrikson et al. [2015] extended MIAs to shallow neural networks for reconstructing low-resolution grayscale face images. While effective for simple models, this method fails when applied to deep neural networks (DNNs) handling high-dimensional data, as reconstructions often lack semantic relevance.

To address these limitations, Zhang et al. [2020] introduced the two-stage *generative model inversion* approach, which leverages generative adversarial networks (GANs) [Goodfellow et al., 2014, Radford et al., 2016] to learn an image prior from public auxiliary datasets and constrains the attack optimization to the generator's latent space. This breakthrough significantly improved the visual quality and semantic fidelity of reconstructed samples and has since fueled major advances in the field of MIAs, particularly for high-dimensional image data [Zhou et al., 2024, Han et al., 2025]. Recent works can be categorized by the model inversion adversary's access level: white-box, black-box, and label-only settings, each posing unique challenges and guiding corresponding defense developments.

In the *white-box* setting, where attackers have full access to the model architecture and weights, most works follow the generative model inversion framework. KEDMI [Chen et al., 2021] enhanced this by introducing an advanced discriminator that incorporates knowledge from the target model. VMI [Wang et al., 2021a] recast the problem as variational inference, using a Bayesian framework to balance diversity and fidelity. PPA [Struppek et al., 2022] further pushed the frontier by leveraging pre-trained StyleGAN generators and introducing the Poincaré loss to replace cross-entropy (CE) loss, addressing gradient vanishing issues in the inversion process. Similarly, Nguyen et al. [2023b] proposed the *logit maximization* (LOM) loss as an alternative to CE loss, alongside model augmentation techniques to mitigate overfitting. PLG-MI [Yuan et al., 2023] advanced MIAs by integrating a conditional GAN (cGAN) with max-margin loss and pseudo-label guidance, effectively decoupling class-specific search spaces and enhancing the exploitation of target model information. These methods primarily concentrate on either the initial training process of GANs or the optimization techniques used in the attacks. A Recent work PPDG-MI [Liu et al., 2020, Peng et al., 2024b] took a different direction by fine-tuning the GAN generator post-attack with reconstructed samples, narrowing the distribution gap between prior and private data distributions.

In the *black-box* setting, where attackers can only query the model, An et al. [2022] introduced a genetic search approach to replace gradient-based optimization, while RLB-MI [Han et al., 2023] framed the attack as a Markov decision process (MDP) and applied reinforcement learning to optimize the latent vector. In the *label-only* setting, which is the most restrictive scenario where only hard labels are available, Kahla et al. [2022] proposed the *boundary-repelling model inversion* (BREP-MI) method, which uses zeroth-Order Optimization method to approximate gradient descent and steer the search toward dense class regions. Inspired by transfer learning, Nguyen et al. [2023a] introduced *label-only via knowledge transfer* (LOKT), which uses a target model-assisted ACGAN (T-ACGAN) to effectively transform the label-only attack into a white-box setting.

Many studies have also focused on designing defense methods against generative MIAs. Since MIAs exploit the strong correlation between inputs and outputs for successful attacks, Wang et al. [2020] proposed augmenting the standard classification objective with a mutual information regularizer to penalize this correlation. However, this approach can significantly degrade the model's predictive performance. To overcome this limitation, Peng et al. [2022] introduced bilateral dependency optimization (BiDO), which enhances the dependency between input features and latent representations while minimizing the dependency between representations and outputs [Peng et al., 2025]. Inspired by BiDO, Stealthy Shield Defense (SSD) [Zhuang et al., 2025] adopts an inference-time strategy that minimizes mutual information between input features and predictions while maximizing the mutual information between predictions and labels, providing an effective black-box defense. Additionally, Ho et al. [2024] proposed freezing the early layers of a pre-trained model and fine-tuning the remaining layers on private data to reduce vulnerability for reconstruction attacks. Struppek et al. [2024] further observed that negative label smoothing can also mitigate generative MIAs. In a recent work, Hao et al. [2024] examined the impact of model architecture on MIA robustness and found that residual connections can increase vulnerability to these attacks.

## B    Derivation of the Alignment-aware Training Objective

In this section, we provide a derivation of the Inequality (6), which serves as a relaxation used to obtain the final alignment-aware training objective in Eq. (7).

**Lemma B.1.** *Under the same notation as in Section 4, and assuming all gradient vectors $\nabla_{\mathbf{x}} f_i(\mathbf{x}; \boldsymbol{\theta})$ have equal norm, the following inequality holds:*

$$-\frac{\left\|\widetilde{\mathbf{P}}_{\mathbf{x}} \sum_{i=1}^{C} \nabla_{\mathbf{x}} f_i(\mathbf{x}; \boldsymbol{\theta})\right\|}{\left\|\sum_{i=1}^{C} \nabla_{\mathbf{x}} f_i(\mathbf{x}; \boldsymbol{\theta})\right\|} \geq -\frac{1}{C} \sum_{i=1}^{C} \frac{\left\|\widetilde{\mathbf{P}}_{\mathbf{x}} \nabla_{\mathbf{x}} f_i(\mathbf{x}; \boldsymbol{\theta})\right\|}{\left\|\nabla_{\mathbf{x}} f_i(\mathbf{x}; \boldsymbol{\theta})\right\|}. \tag{9}$$

*Proof.* Let $g_i := \nabla_{\mathbf{x}} f_i(\mathbf{x}; \boldsymbol{\theta})$ and assume $\|g_i\| = a > 0$ for all $i$. Put $g := \sum_{i=1}^{C} g_i$ and suppose $g \neq 0$. By linearity of the orthogonal projector $\widetilde{\mathbf{P}}_{\mathbf{x}}$,

$$\widetilde{\mathbf{P}}_{\mathbf{x}} g = \sum_{i=1}^{C} \widetilde{\mathbf{P}}_{\mathbf{x}} g_i.$$

Applying the triangle inequality,

$$\left\|\widetilde{\mathbf{P}}_{\mathbf{x}} g\right\| = \left\|\sum_{i=1}^{C} \widetilde{\mathbf{P}}_{\mathbf{x}} g_i\right\| \leq \sum_{i=1}^{C} \left\|\widetilde{\mathbf{P}}_{\mathbf{x}} g_i\right\|.$$

Dividing both sides by $\|g\|$ and multiplying by $-1$ reverses the inequality:

$$-\frac{\left\|\widetilde{\mathbf{P}}_{\mathbf{x}} g\right\|}{\|g\|} \geq -\frac{\sum_{i=1}^{C} \left\|\widetilde{\mathbf{P}}_{\mathbf{x}} g_i\right\|}{\|g\|}. \tag{10}$$

Since $\|g\| = \left\|\sum_{i=1}^{C} g_i\right\| \leq \sum_{i=1}^{C} \|g_i\| = Ca$, we have $\frac{1}{\|g\|} \geq \frac{1}{Ca}$. Substituting this bound into the right-hand side of Inequality (10) yields

$$-\frac{\sum_{i=1}^{C} \left\|\widetilde{\mathbf{P}}_{\mathbf{x}} g_i\right\|}{\|g\|} \geq -\frac{\sum_{i=1}^{C} \left\|\widetilde{\mathbf{P}}_{\mathbf{x}} g_i\right\|}{Ca} = -\frac{1}{C} \sum_{i=1}^{C} \frac{\left\|\widetilde{\mathbf{P}}_{\mathbf{x}} g_i\right\|}{a}. \tag{11}$$

Combining Inequalities (10) and (11) and substituting $a = \|g_i\|$ gives

$$-\frac{\left\|\widetilde{\mathbf{P}}_{\mathbf{x}} g\right\|}{\|g\|} \geq -\frac{1}{C} \sum_{i=1}^{C} \frac{\left\|\widetilde{\mathbf{P}}_{\mathbf{x}} g_i\right\|}{\|g_i\|},$$

which is exactly Inequality (9). Equality holds iff both triangle inequalities above are tight, *i.e.,* (i) all vectors $\widetilde{\mathbf{P}}_{\mathbf{x}} g_i$ are colinear and (ii) all $g_i$ themselves are colinear. $\qquad\square$

## C    Algorithmic Realizations of Gradient–Manifold Alignment Methods

This section presents the algorithmic implementations of our proposed training objective for validating the hypothesis, as well as the training-free alignment approach designed to enhance gradient–manifold alignment and improve model inversion performance.

**(1) Alignment-Aware Training.** To validate our hypothesis that stronger alignment between loss gradients and the generator manifold leads to greater inversion vulnerability, we introduce a gradient–manifold alignment-aware training objective. This objective augments the standard classification loss with a geometric alignment term and can be optimized via standard backpropagation. The training procedure is detailed in Algorithm 1.

**(2) Training-Free Alignment Promotion.** Motivated by the above findings, we propose a training-free method that improves gradient–manifold alignment at inversion time. By averaging loss gradients over perturbed or transformed versions of the synthetic input, this approach denoises the gradient signal in a geometry-aware manner. The inference-time procedure is described in Algorithm 2.

**Algorithm 1** Gradient–Manifold Alignment-Aware Training
___
**Input:** Classifier $f(\cdot\,; \boldsymbol{\theta})$, pre-trained VAE decoder $\mathcal{D}$, training set $\mathcal{D}_{\text{pri}}$, trade-off hyperparameter $\beta$, number of training steps $T$
**Output:** Updated target model parameters $\boldsymbol{\theta}$
___
 1: **for** $t = 1$ to $T$ **do**
 2:    Sample a minibatch $\{(\mathbf{x}^{(n)}, y^{(n)})\}_{n=1}^{B}$ from $\mathcal{D}_{\text{pri}}$
 3:    **for** each $(\mathbf{x}, y)$ in batch **do**
 4:        Compute latent code: $\mathbf{z} \leftarrow \mathcal{E}(\mathbf{x})$
 5:        Compute Jacobian: $J_{\mathcal{D}}(\mathbf{z}) = \frac{\partial \mathcal{D}}{\partial \mathbf{z}}$
 6:        Compute SVD: $J_{\mathcal{D}}(\mathbf{z}) = \mathbf{U}\boldsymbol{\Sigma}\mathbf{V}^{\top}$
 7:        Let $\mathbf{U}_k$ be the first $k$ columns of $\mathbf{U}$
 8:        Estimate projection matrix: $\widetilde{\mathbf{P}}_{\mathbf{x}} \leftarrow \mathbf{U}_k \mathbf{U}_k^{\top}$
 9:        Compute softmax probabilities: $p = \text{softmax}(f(\mathbf{x}; \boldsymbol{\theta}))$
10:        Compute CE loss: $\mathcal{L}_{\text{CE}} = -\log p_y$
11:        Compute input gradients of logits: $\{\nabla_{\mathbf{x}} f_i(\mathbf{x}; \boldsymbol{\theta})\}_{i=1}^{C}$
12:        Compute gradient sum: $\mathbf{g} = \sum_{i=1}^{C} \nabla_{\mathbf{x}} f_i(\mathbf{x}; \boldsymbol{\theta})$
13:        Compute alignment term: $\mathcal{L}_{\text{align}}^{\text{geo}} \leftarrow \frac{\|\widetilde{\mathbf{P}}_{\mathbf{x}} \mathbf{g}\|}{\|\mathbf{g}\|}$
14:        Compute final loss: $\mathcal{L}_{\text{align}}(\boldsymbol{\theta}) \leftarrow \mathcal{L}_{\text{CE}} - \beta \cdot \mathcal{L}_{\text{align}}^{\text{geo}}$
15:    **end for**
16:    Update $\boldsymbol{\theta}$ via backpropagation over average batch loss
17: **end for**
18: **return** $\boldsymbol{\theta}$
___

**Algorithm 2** Training-Free Gradient–Manifold Alignment During Inversion
___
**Input:** Target model $f$, pre-trained generator G, inversion loss $\mathcal{L}$, initial latent code $\mathbf{z}$, number of inversion steps $T$, number of samples $K$, perturbation strength $\alpha$, sampling strategy $\rho \in \{\texttt{PAA}, \texttt{TAA}\}$
**Output:** Recovered image $\widehat{\mathbf{x}} = \text{G}(\mathbf{z})$
___
 1: **for** $t = 1$ to $T$ **do**
 2:    $\mathbf{x} \leftarrow \text{G}(\mathbf{z})$
 3:    Initialize gradient buffer: $\mathcal{G} \leftarrow \emptyset$
 4:    **for** $k = 1$ to $K$ **do**
 5:        **if** $\rho = \texttt{PAA}$ **then**
 6:            Compute noise scale: $\sigma \leftarrow \alpha \left(\max(\mathbf{x}) - \min(\mathbf{x})\right)$
 7:            Sample noise: $\boldsymbol{\epsilon}_k \sim \mathcal{N}(0, \sigma^2 \mathbf{I})$
 8:            $\mathbf{x}_k \leftarrow \mathbf{x} + \boldsymbol{\epsilon}_k$
 9:        **else if** $\rho = \texttt{TAA}$ **then**
10:            Sample transformation: $\tau_k \sim \mathcal{T}$
11:            $\mathbf{x}_k \leftarrow \tau_k(\mathbf{x})$
12:        **end if**
13:        Compute loss gradient: $\mathbf{g}_k \leftarrow \nabla_{\mathbf{x}_k} \mathcal{L}(\mathbf{x}_k)$
14:        Append to buffer: $\mathcal{G} \leftarrow \mathcal{G} \cup \{\mathbf{g}_k\}$
15:    **end for**
16:    Compute averaged gradient: $\widetilde{\nabla}\mathcal{L}(\mathbf{x}) \leftarrow \frac{1}{K} \sum_{k=1}^{K} \mathbf{g}_k$
17:    Update latent code: $\mathbf{z} \leftarrow \mathbf{z} - \eta\, J_{\text{G}}(\mathbf{z})^{\top} \widetilde{\nabla}\mathcal{L}(\mathbf{x})$
18: **end for**
19: **return** $\widehat{\mathbf{x}} = \text{G}(\mathbf{z})$
___

# D   Experimental Setup and Implementation Details

## D.1   Hard- and Software Details

All high-resolution MIA experiments using *Plug & Play Attacks* (PPA) were conducted on Oracle Linux Server 8.9 with NVIDIA A100-80G GPUs, using CUDA 11.7, Python 3.9.18, and PyTorch

Table 3: A summary of experimental setups.

| Setting | MIAs | Private Dataset | Public Dataset | Target Model | Evaluation Model |
|---|---|---|---|---|---|
| Low-resolution setting | GMI (LOMMA) / KEDMI (LOMMA) / PLG-MI | CelebA | CelebA / FFHQ | VGG16 / FaceNet (64) | FaceNet (112) |
| High-resolution setting | PPA | CelebA / FaceScrub | FFHQ | ResNet-18 / DenseNet-121 / ResNeSt-50 | Inception-v3 |

1.13.1. Low-resolution facial recognition MIAs were run on Ubuntu 20.04.4 LTS with NVIDIA RTX 3090 GPUs, under CUDA 11.6, Python 3.7.12, and PyTorch 1.13.1.

## D.2 Target Models

**(1) Empirical Validation of the Hypothesis.** To validate our hypothesis, we conduct experiments on models pre-trained for a 1000-class classification task using $64 \times 64$ CelebA images. The model and training pipeline are based on the implementation provided at `https://github.com/sutd-visual-computing-group/Re-thinking_MI`. To compute alignment scores, we require estimates of the tangent space at each training point. These are obtained using a pre-trained VAE decoder, which maps latent representations back to the image space. For each training image, we compute the Jacobian of the decoder to extract the local tangent basis and pre-store it for downstream alignment computation. However, this procedure is memory-intensive. For example, estimating and storing tangent bases for approximately $2,700$ training images from the first 100 classes of CelebA requires about 30 GB of disk space. Due to this storage constraint and the exploratory nature of the analysis, we restrict our investigation to a 100-class subset of the full dataset.

To obtain models trained on this 100-class subset, we first adapt the original 1000-class model by fine-tuning it on the corresponding subset. Fine-tuning is performed for 20 epochs using stochastic gradient descent with an initial learning rate of $10^{-2}$, momentum of 0.9, weight decay of $10^{-4}$, and batch size of 128. The learning rate is scheduled to decrease by a factor of 0.02 at epochs 10 and 15. This procedure yields a 100-class vanilla model. Subsequently, to obtain models with varying levels of training-time gradient–manifold alignment, we continue fine-tuning the 100-class vanilla model for 30 additional epochs using our proposed alignment-aware training objective. The learning rate is fixed throughout this phase. To capture the evolution of training-time alignment scores, we save model checkpoints at intermediate epochs. These models serve as the basis for evaluating the correlation between alignment and model inversion vulnerability in later experiments.

**(2) Evaluation of Proposed Methods.** To evaluate our proposed methods, we adopt distinct training configurations for models at different image resolutions. For high-resolution inputs ($224 \times 224$) from the CelebA and FaceScrub datasets, we follow the setup from Struppek et al. [2022]. Models are optimized using Adam [Kingma and Ba, 2015] with an initial learning rate of $10^{-3}$, $\beta$ parameters set to $(0.9, 0.999)$, and a weight decay of $10^{-3}$. Training runs for 100 epochs with a batch size of 128, and the learning rate is reduced by a factor of 0.1 at epochs 75 and 90. Input preprocessing includes normalization (mean and standard deviation both set to 0.5), followed by a sequence of augmentations: random cropping with a scale range of $[0.85, 1.0]$ and fixed aspect ratio of 1.0, resizing to $224 \times 224$, and horizontal flipping with a probability of 0.5.

For low-resolution images ($64 \times 64$) from CelebA, we follow the training protocol provided by `https://github.com/sutd-visual-computing-group/Re-thinking_MI`. Specifically, we use stochastic gradient descent (SGD) with an initial learning rate of $10^{-2}$, momentum of 0.9, and weight decay of $10^{-4}$. Models are trained for 100 epochs with a batch size of 64, and the learning rate is decayed by a factor of 0.1 at epochs 75 and 90.

## D.3 Evaluation Models

For our PPA-based experiments, we follow the original implementation at `https://github.com/LukasStruppek/Plug-and-Play-Attacks` to train Inception-v3 evaluation models, using the training configurations specified in Struppek et al. [2022]. These models achieve test accuracies of

96.53% on FaceScrub and 94.87% on CelebA. To compute K-nearest neighbor (KNN) distances, which serve as a similarity metric between reconstructed and true samples in facial recognition tasks, we adopt the pre-trained FaceNet model [Schroff et al., 2015], available at `https://github.com/timesler/facenet-pytorch`.

For experiments on target models trained on $64 \times 64$ resolution CelebA dataset, we use an evaluation model from `https://github.com/sutd-visual-computing-group/Re-thinking_MI`. This model is based on the face.evoLVe architecture [Wang et al., 2021b] with a modified ResNet-50 backbone, and achieves a reported test accuracy of 95.88%. Details on the training procedure are available in Zhang et al. [2020].

### D.4 Attack Parameters

**High-Resolution Setting.** In the high-resolution setting, we follow the *Plug & Play Attack* (PPA) method, which comprises three stages: (1) latent code pre-selection, (2) latent code optimization, and (3) result selection. During pre-selection, we sample 2000 latent codes per class and retain the top 100 candidates based on the target model's response for both CelebA and FaceScrub datasets. In the optimization stage, we perform 70 iterations of gradient-based latent code updates per class. The final result selection stage is omitted in our implementation in order to include as many as samples for evaluation. We focus on the first 100 classes, generating 100 reconstructed samples per class.

As for the parameters of PAA strategy, we use Gaussian perturbations of standard deviation $\sigma$ set to 5% of the synthesized images' dynamic range. For parameters of TAA strategy, we apply three geometrically constrained transformations: random resized cropping with scale factors spanning $[0.8, 1.0]$ and aspect ratios limited to $[0.9, 1.1]$, horizontal flipping with probability $p = 0.5$, and random rotations within $\pm 5°$ angular displacement.

**Low-Resolution Setting.** In the low-resolution setting, we target the first 100 classes from CelebA as the private dataset $\mathcal{D}_{\text{pri}}$ and generate 100 samples per identity using CelebA, FFHQ and FaceScrub as auxiliary datasets $\mathcal{D}_{\text{aux}}$. For instantiations of AlignMI, we maintain identical PAA and TAA parameter configurations from the high-resolution setup unless explicitly stated. Implementation details differ slightly across MIAs. For GMI (LOMMA) using StyleGAN, we directly sample and optimize 100 latent codes for 100 steps with a batch size of 20, and set the PAA's Gaussian noise standard deviation $\sigma$ is set to 0.5% of the synthesized images' dynamic range. For KEDMI (LOMMA) with DCGAN, we process 100 samples per identity through 200 optimization steps with a batch size of 100. For PLG-MI with a cGAN prior, the baseline includes a data augmentation pipeline comprising: random resized cropping to $64 \times 64$ with scale in $[0.8, 1.0]$ and fixed aspect ratio 1.0, color jittering with brightness and contrast set to $\pm 0.2$, random horizontal flips (probability 0.5), and rotations within $\pm 5°$. In our PAA and TAA configurations, we omit this augmentation pipeline to isolate the effect of gradient–manifold alignment. Optimization for PLG-MI runs for 100 steps with a batch size of 20.

Due to the high computational cost of generative MIAs, we perform a single attack per target model. To reduce randomness, we generate at least 100 inversion samples per class across all configurations.

### D.5 Evaluation Metrics

**Attack Accuracy (Attack Acc).** We employ an evaluation model (generally more robust and powerful than the target model) trained on the same dataset as the target model to verify whether reconstructed images correctly represent the target class, following the evaluation method of Zhang et al. [2020]. This metric serves as an automated proxy for human evaluation, assessing how well the reconstructed images capture the distinctive characteristics of the target class compared to other classes. The attack accuracy is computed as the percentage of predictions matching the target class, reporting both top-1 (Acc@1) and top-5 (Acc@5) accuracy scores.

**K-Nearest Neighbors Distance (KNN Dist).** KNN distance quantifies reconstruction quality through $l_2$ distance computation in a model's feature embedding space, measuring the similarity between reconstructed images and their nearest original private training samples. This metric serves as a quantitative indicator of visual fidelity, where smaller distances correspond to higher similarity between generated and genuine training data. For high-resolution attacks in PPA [Struppek et al., 2022], we extract features from FaceNet's penultimate layer [Schroff et al., 2015], while for low-resolution model inversion attacks, we use the evaluation model's penultimate layer features.

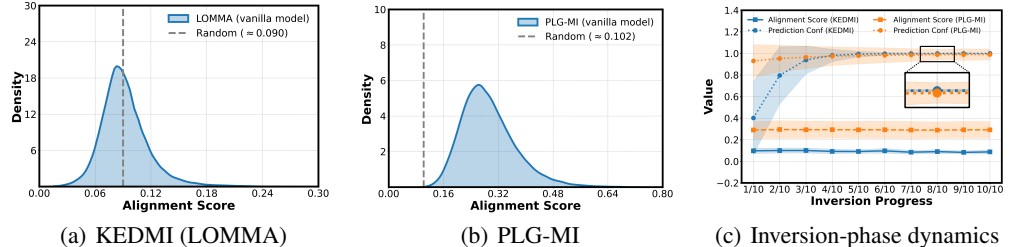

| (a) KEDMI (LOMMA) | (b) PLG-MI | (c) Inversion-phase dynamics |

Figure 6: **Additional gradient-manifold alignment during inversion process.** **(a)** Alignment score distribution for KEDMI (LOMMA) using an inversion-specific GAN trained on CelebA. **(b)** Corresponding results for PLG-MI using a conditional GAN. **(c)** Evolution of mean alignment scores versus prediction confidence during inversion. Notably, while prediction confidence demonstrates monotonic improvement throughout the inversion process, gradient-manifold alignment in additional attack methods also remains stable and low, reinforcing the lack of correlation between confidence and gradient–manifold alignment.

## D.6 Experimental Details for Figure 3

**Low-Resolution Setting.** In the low-resolution experiments, we adopt a DCGAN trained on CelebA as the generative prior. The latent space dimension of DCGAN is 100, corresponding to a random baseline alignment score of approximately 0.090. The target classifier is a VGG16 model trained on CelebA, and the inversion targets the first 25 classes, each containing $1,000$ images. For Fig. 3(a), we run the inversion optimization for $1,200$ steps and record the inversion-time alignment scores of the loss gradients every 10 steps for each reconstructed sample. The figure presents the distribution of all collected alignment scores. In Fig. 3(c), we further analyze temporal dynamics by averaging alignment scores across all classes at each step, illustrating how gradient–manifold alignment evolves during optimization.

Additionally, we evaluate gradient–manifold alignment during the inversion process for other attack methods, including KEDMI (LOMMA) and PLG-MI, in the low-resolution setting. The results are present in Fig. 6. Both methods leverage CelebA as the generative prior and target a VGG16 classifier trained on CelebA. Specifically for KEDMI, we adopt a DCGAN with latent space dimension of DCGAN 100, corresponding to a random baseline alignment score approximately 0.090. The inversion process targets the first 50 classes, each containing 500 images and proceeds $1,200$ optimization steps. For PLG-MI, we use a conditional GAN (cGAN) with 128 latent dimensions, which corresponds to a random baseline alignment score approximately 0.102. The inversion process executes 100 optimization iterations targeting the first 100 classes, each containing 100 images.

Interestingly, the PLG-MI method exhibits higher inversion-time alignment scores than GMI (LOM) and KEDMI (LOM). This improvement can be attributed to its use of a conditional GAN, which incorporates label information throughout the inversion process. The stronger alignment may partially explain PLG-MI's superior attack performance.

**High-Resolution Setting.** In the high-resolution experiments, we use a StyleGAN model trained on FFHQ as the generative prior. The latent space has dimension 512, yielding a random baseline alignment score of approximately 0.058. The target classifier is a ResNet18 model trained on CelebA, with inversion targeting the first 50 classes, each containing 50 images. For Fig. 3(b), inversion is run for 100 steps, with alignment scores recorded at 10 equally spaced intervals per reconstructed sample. The figure shows the distribution of the recorded scores. In Fig. 3(c), we track temporal alignment by averaging scores over all latent vectors at each interval, capturing how alignment develops throughout the inversion process.

## D.7 Experimental Details for Figure 4

**Tangent Space Estimation.** To compute training-time alignment scores, we estimate the tangent space at each training sample using a pre-trained VAE from Stable Diffusion. Specifically, the VAE encoder maps an input image $\mathbf{x}$ of shape $64 \times 64 \times 3$ to a latent representation $\mathbf{z}$ of shape $8 \times 8 \times 4$, which is then decoded back to the image space by the VAE decoder. For each training image, we compute the Jacobian of the decoder to obtain the local tangent basis, resulting in a Jacobian matrix of

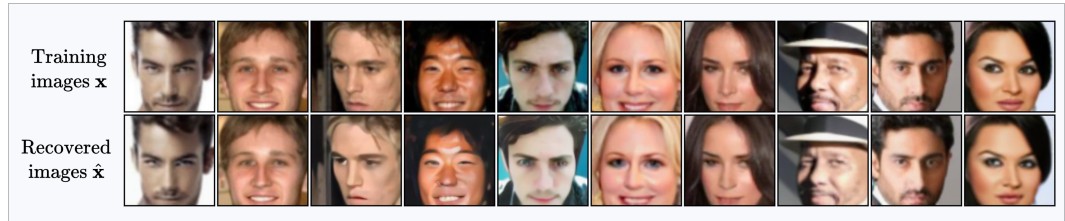

Figure 7: Original training samples (top row) and corresponding reconstructions (bottom row) from the pre-trained VAE used for tangent space estimation. The visual similarity confirms the VAE's ability to approximate the natural image manifold reliably.

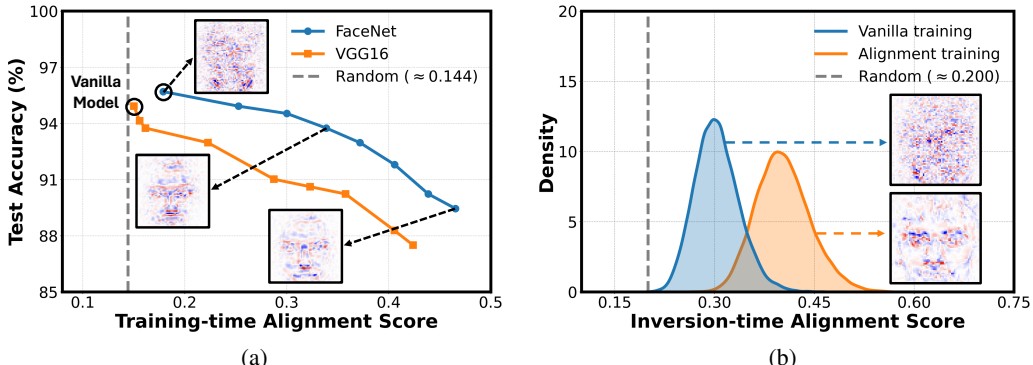

(a)             (b)

Figure 8: **Empirical evaluation of gradient–manifold alignment (enlarged version). (a)** Test accuracy vs. training-time alignment score ($AS_{tr}$) for models sampled during fine-tuning vanilla models with the alignment-aware training objective. Insets show input gradient visualizations for models with varying degrees of alignment. **(b)** Distribution of inversion-time alignment scores ($AS_{inv}$) for the vanilla model compared to the alignment-aware model.

shape $12{,}288 \times 256$. This process is memory-intensive: for example, estimating and storing tangent bases for approximately 2,700 training samples from the first 100 classes of CelebA consumes roughly 30 GB of disk space. As shown in Fig. 7, the reconstructed images closely match the original inputs, indicating that the pre-trained VAE, despite not being trained on the target dataset, offers a reliable approximation of the natural image manifold.

**Empirical evaluation of gradient-manifold alignment.** To empirically evaluate the trade-off between test accuracy and training-time alignment score as shown in Fig. 4(a) (or Fig. 8(a)), we conducted experiments using two 100-class target models: VGG16 and FaceNet. The training procedures for these models followed the same specifications detailed in Appendix D.2. During training, we saved intermediate model checkpoints at various epochs to capture the evolution of model performance under our alignment-aware objective.

For analyzing the distribution of inversion-time alignment scores presented in Fig. 4(b) (or Fig. 8(b)), we select two 100-class FaceNet models as target models. The vanilla model achieves a test accuracy of 96.53% with training-time alignment score $AS_{tr} = 0.175$, while the aligned model achieves a test accuracy of 93.75% with $AS_{tr} = 0.339$. We use the GMI (LOM) attack method with StyleGAN as a prior, targeting the first 25 classes and running the optimization for 100 steps with batch size 20 for both the vanilla and aligned models.

In Fig. 4(c), we extend our evaluation to 1000-class VGG16 models, following the same training protocol as described in Appendix D.2. We save checkpoints at intermediate training epochs to obtain models with varying test accuracies. The alignment scores $AS_{tr}$ are recorded throughout the training process. Additionally, we compute the alignment scores $AS_{inv}$ using the GMI (LOM) attack with StyleGAN, again targeting the first 25 classes and running the optimization for 100 steps.

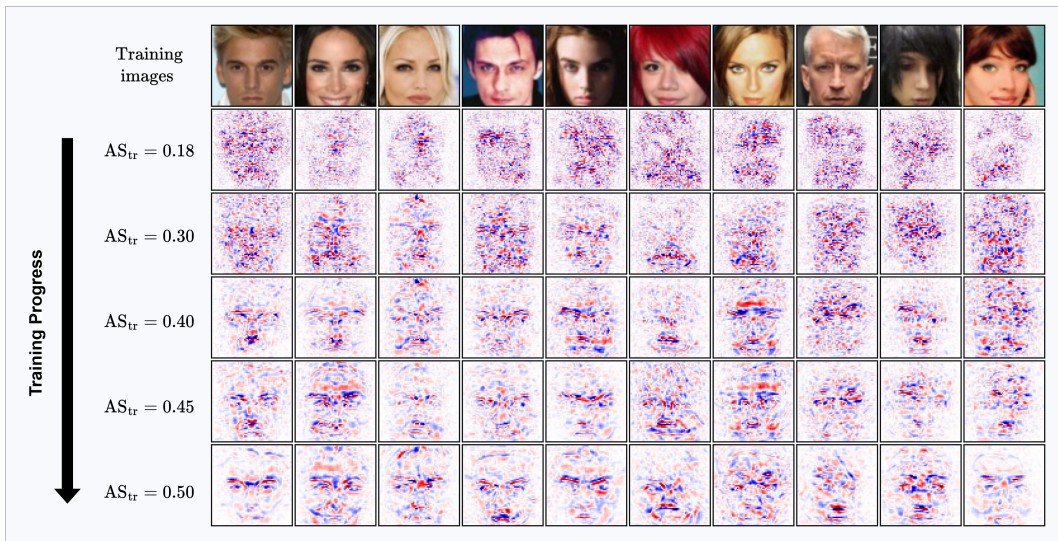

Figure 9: **Training-time alignment progression with alignment-aware training.** Evolution of training-time alignment score ($AS_{tr}$) and gradient visualizations during fine-tuning of FaceNet using our alignment-aware objective. As alignment improves, loss gradients exhibit increasingly structured and semantically meaningful patterns. (Best viewed with zoom.)

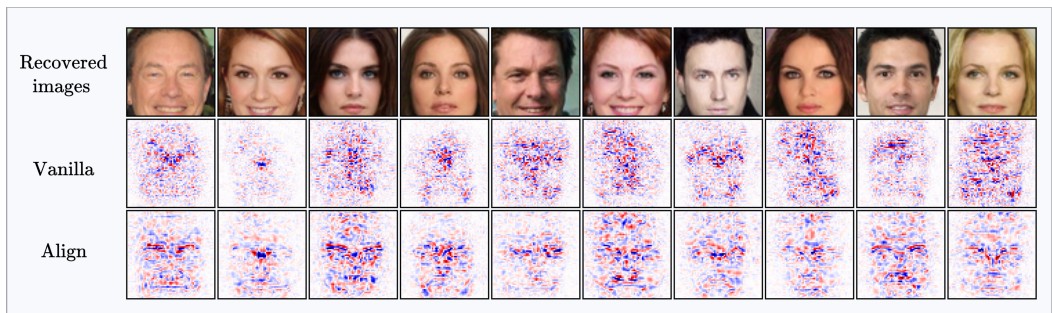

Figure 10: **Comparison of inversion-time loss gradients.** Visualization of loss gradients from the vanilla model (top) and the alignment-aware model (bottom). The alignment-aware model produces gradients that are sharper and more semantically aligned with facial structures, indicating stronger alignment with the generator manifold. (Best viewed with zoom.)

# E  Additional Experimental Results

## E.1  Additional Empirical Validation of the Hypothesis

We illustrate the fine-tuning progress of a FaceNet model optimized with our alignment-aware objective in Fig. 10. As fine-tuning proceeds, the training-time alignment score ($AS_{tr}$) consistently increases, and corresponding gradient visualizations exhibit progressively clearer and more semantically meaningful structures. This demonstrates the effectiveness of our alignment-aware training strategy in promoting geometrically informative gradients.

For comparison, Fig. 10 also presents inversion-time loss gradient images from both the vanilla and alignment-aware models. The gradients from the alignment-aware model reveal clearer, semantically meaningful structures, highlighting improved alignment with the underlying generator manifold.

To further validate our hypothesis, we extend our experiments to include IR152 as the target model, using the GMI (LOM) attack method. As shown in Fig. 11(a), the results are consistent with our earlier findings in Fig. 4(a) (Sec. 6.2): as fine-tuning progresses, the training-time alignment score ($AS_{tr}$) steadily increases, and corresponding gradient visualizations reveal increasingly semantically

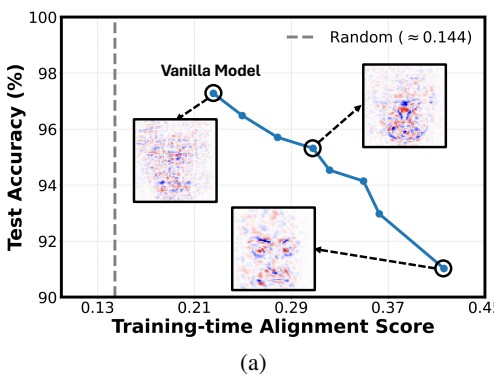 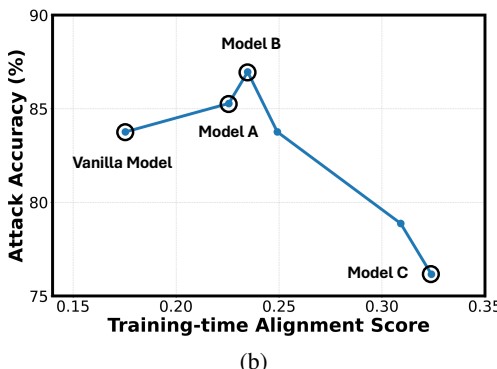

|  (a)  |  (b)  |

Figure 11: **Additional empirical evaluation of gradient–manifold alignment. (a)** Test accuracy vs. training-time alignment score ($AS_{tr}$) for IR152 models sampled during fine-tuning vanilla models with the alignment-aware training objective. Insets show input gradient visualizations for models with varying degrees of alignment. **(b)** MIA success on vanilla and alignment-aware IR152 models with different $AS_{tr}$.

Table 4: Comparison of inversion performance with PPA in the high-resolution setting. $\mathcal{D}_{pri}$ = CelebA or FaceScrub, GANs are pre-trained on $\mathcal{D}_{aux}$ = FFHQ. The symbol ↓ (or ↑) indicates that smaller (or larger) values are preferred, and the green numbers represent the performance improvement. The results are averaged over three independent runs.

| Target Model | Method | CelebA | | | FaceScrub | | |
|---|---|---|---|---|---|---|---|
| | | Acc@1↑ | Acc@5↑ | KNN Dist↓ | Acc@1↑ | Acc@5↑ | KNN Dist↓ |
| **ResNet-18** | PPA | $85.63 \pm 1.39$ | $95.12 \pm 0.80$ | $0.693 \pm 0.009$ | $81.57 \pm 0.25$ | $94.85 \pm 0.05$ | $0.796 \pm 0.003$ |
| | + PAA (ours) | $88.75 \pm 1.63$ | $96.59 \pm 0.75$ | $0.669 \pm 0.006$ | $83.97 \pm 0.29$ | $95.78 \pm 0.14$ | $0.777 \pm 0.003$ |
| | + TAA (ours) | $91.68 \pm 0.19$ | $97.68 \pm 0.04$ | $0.662 \pm 0.001$ | $93.68 \pm 0.05$ | $98.84 \pm 0.08$ | $0.691 \pm 0.001$ |
| **DenseNet-121** | PPA | $82.22 \pm 0.44$ | $93.26 \pm 0.39$ | $0.708 \pm 0.002$ | $75.66 \pm 0.46$ | $90.91 \pm 0.22$ | $0.786 \pm 0.002$ |
| | + PAA (ours) | $85.64 \pm 0.15$ | $95.16 \pm 0.50$ | $0.684 \pm 0.003$ | $80.70 \pm 0.22$ | $93.40 \pm 0.09$ | $0.761 \pm 0.003$ |
| | + TAA (ours) | $87.88 \pm 0.86$ | $96.20 \pm 0.41$ | $0.687 \pm 0.008$ | $86.54 \pm 0.73$ | $95.12 \pm 0.52$ | $0.712 \pm 0.004$ |
| **ResNeSt-50** | PPA | $70.75 \pm 0.41$ | $87.43 \pm 0.36$ | $0.793 \pm 0.001$ | $71.58 \pm 0.19$ | $90.60 \pm 0.28$ | $0.827 \pm 0.004$ |
| | + PAA (ours) | $75.71 \pm 0.06$ | $90.48 \pm 0.09$ | $0.764 \pm 0.002$ | $73.38 \pm 0.18$ | $91.34 \pm 0.16$ | $0.807 \pm 0.004$ |
| | + TAA (ours) | $79.19 \pm 0.63$ | $92.28 \pm 0.12$ | $0.761 \pm 0.002$ | $84.38 \pm 0.41$ | $96.04 \pm 0.21$ | $0.753 \pm 0.002$ |

Table 5: Comparison of inversion performance with white-box MIAs in the low-resolution setting. Target model $f$ = VGG16 trained on $\mathcal{D}_{pri}$ = CelebA. GANs are trained on $\mathcal{D}_{aux}$ = CelebA or FFHQ.

| Method | CelebA | | | | FFHQ | | | |
|---|---|---|---|---|---|---|---|---|
| | Acc@1↑ | Acc@5↑ | KNN Dist↓ | Ratio↓ | Acc@1↑ | Acc@5↑ | KNN Dist↓ | Ratio↓ |
| GMI (LOMMA) | 94.12 | 98.93 | 1155.02 | / | 73.07 | 92.95 | 1288.08 | / |
| + PAA (ours) | 94.65 (+0.53) | 99.00 (+0.07) | 1104.52 (-50.50) | 10.79 | 72.11 (-0.96) | 92.55 (-0.40) | 1292.79 (+4.71) | 11.11 |
| + TAA (ours) | 96.36 (+2.24) | 99.44 (+0.51) | 1105.63 (-49.38) | 4.76 | 81.25 (+8.18) | 96.02 (+3.07) | 1255.01 (-33.07) | 12.38 |
| KEDMI (LOMMA) | 60.46 | 87.35 | 1275.10 | / | 26.32 | 52.65 | 1592.32 | / |
| + PAA (ours) | 76.75 (+16.29) | 95.55 (+8.20) | 1266.46 (-8.65) | 14.72 | 25.86 (-0.46) | 52.74 (+0.09) | 1595.91 (+3.59) | 17.27 |
| + TAA (ours) | 59.67 (-0.79) | 86.83 (-0.52) | 1364.61 (+89.51) | 9.33 | 26.12 (-0.20) | 52.95 (+0.30) | 1595.83 (+3.51) | 18.42 |

meaningful features. Notably, this rise in alignment is accompanied by a gradual decline in test accuracy, reaffirming the trade-off between alignment and generalization.

Additionally, we evaluate model inversion performance across both vanilla and alignment-aware models with varying levels of $AS_{tr}$. As shown in Fig. 11(b), the trend mirrors Fig. 5: MIA vulnerability increases with alignment up to a certain threshold, after which further increases in $AS_{tr}$ reduce attack success. This characteristic inverted V-shaped relationship supports our hypothesis and demonstrates that the correlation between gradient–manifold alignment and inversion vulnerability holds across different model architectures.

### E.2 Additional Evaluations of Proposed Methods

**Inversion-Time Alignment Score Comparison with PPA in High-Resolution Setting.** Fig.12 shows the distribution of inversion-time alignment scores for the baseline method and our training-free variants, PAA and TAA. These results are obtained using the PPA attack on a ResNet-18 model trained

Table 6: Comparison of inversion performance with white-box MIAs in the low-resolution setting. Target model $f$ = FaceNet trained on $\mathcal{D}_{pri}$ = CelebA. GANs are trained on $\mathcal{D}_{aux}$ = CelebA or FFHQ.

| Method | CelebA | | | | FFHQ | | | |
|---|---|---|---|---|---|---|---|---|
| | Acc@1↑ | Acc@5↑ | KNN Dist↓ | Ratio↓ | Acc@1↑ | Acc@5↑ | KNN Dist↓ | Ratio↓ |
| GMI (LOMMA) | 93.66 | 98.25 | 1084.60 | / | 74.01 | 92.91 | 1279.53 | / |
| + PAA (ours) | 93.73 (+0.07) | 98.31 (+0.06) | 1082.41 (-2.19) | 11.64 | 74.36 (+0.35) | 93.22 (+0.31) | 1278.74 (-0.79) | 11.27 |
| + TAA (ours) | 96.74 (+3.08) | 99.11 (+0.86) | 1077.23 (-7.37) | 12.60 | 84.90 (+10.89) | 96.64 (+3.73) | 1234.00 (-45.53) | 15.20 |
| KEDMI (LOMMA) | 60.42 | 89.47 | 1331.94 | / | 30.33 | 61.20 | 1542.77 | / |
| + PAA (ours) | 61.20 (+0.78) | 89.50 (+0.03) | 1342.33 (+10.39) | 15.15 | 29.29 (-1.04) | 61.05 (-0.15) | 1540.16 (-2.61) | 16.13 |
| + TAA (ours) | 60.55 (+0.13) | 89.50 (+0.03) | 1336.16 (+4.22) | 14.73 | 30.28 (-0.05) | 61.43 (+0.23) | 1540.33 (-2.44) | 15.81 |

Table 7: Comparison of inversion performance with PLG-MI in the low-resolution setting. Target model $f$ = FaceNet trained on $\mathcal{D}_{pri}$ = CelebA. GANs are trained on $\mathcal{D}_{aux}$ = FaceScrub or FFHQ.

| Method | FaceScrub | | | | FFHQ | | | |
|---|---|---|---|---|---|---|---|---|
| | Acc@1↑ | Acc@5↑ | KNN Dist↓ | Ratio↓ | Acc@1↑ | Acc@5↑ | KNN Dist↓ | Ratio↓ |
| PLG | 32.06 | 58.17 | 1558.26 | / | 88.68 | 97.06 | 1267.12 | / |
| + PAA (ours) | 29.93 (-2.13) | 53.99 | 1557.11 (-1.15) | 9.07 | 87.32 (-1.36) | 96.37 | 1270.54 (+3.42) | 9.04 |
| + TAA (ours) | 35.99 (+3.93) | 62.87 | 1539.27 (-18.99) | 11.07 | 90.79 (+2.11) | 97.56 | 1256.07 (-11.05) | 11.07 |

on CelebA, with a StyleGAN generator pre-trained on FFHQ. Both PAA and TAA significantly shift the alignment score distribution to the right compared to the vanilla baseline, indicating stronger alignment between the loss gradients and the generator manifold. This enhanced alignment aligns well with the improved gradient visualizations shown in Fig. 13.

**Comparison with white-box MIAs in the low-resolution setting.** In this experiment, we evaluate the performance of two target models, namely VGG16 and FaceNet, under three attack methods: GMI (LOMMA), KEDMI (LOMMA), and PLG-MI. Quantitative results are presented in Tabs. 5, 6, and 7. Overall, AlignMI consistently outperforms baseline methods in most setups, achieving gains in both attack accuracy and KNN distance across different auxiliary datasets. For example, when attacking VGG16 using GMI (LOMMA), PAA increases top-1 accuracy from $94.12\%$ to $94.65\%$ on CelebA, while TAA achieves an additional $2.54\%$ improvement and reduces the KNN distance from $1155.02$ to $1105.63$. Similar trends are observed for KEDMI (LOMMA) and PLG-MI, demonstrating the broad effectiveness of our proposed techniques. However, we also observe occasional performance drops, particularly with PAA in certain KEDMI (LOMMA) and PLG-MI scenarios. This degradation likely arises from the poor visual quality of reconstructions produced by certain low-resolution attacks, especially under significant distribution shifts between the private and public auxiliary datasets. In such cases, additional perturbations further compromise image fidelity, diminishing the effectiveness of neighborhood sampling. As a result, the derived gradients become less informative, leading to occasional failures in inversion.

**Comparisons under SOTA MIA defenses.** Our evaluation focuses on the high-resolution setting, where we assess the effectiveness of our proposed training-free alignment enhancement methods, PAA and TAA, when integrated with state-of-the-art (SOTA) generative model inversion attacks against leading MIA defenses, including BiDO-HSIC [Peng et al., 2022], NegLS [Struppek et al., 2024], and TL-DMI [Ho et al., 2024]. The results, summarized in Tab. 8, show that both PAA and TAA improve inversion performance across all defense scenarios, with TAA consistently achieving the strongest results. All attacks are conducted using the *Plug & Play Attack* (PPA) method, targeting a ResNet-152 classifier trained on $\mathcal{D}_{pri}$ = FaceScrub, with the generative prior provided by a StyleGAN model trained on $\mathcal{D}_{aux}$ = FFHQ. Detailed results are shown in Tab. 8.

For the BiDO-HSIC defense, the baseline inversion performance drops significantly, with top-1 accuracy (Acc@1) of $35.11\%$, top-5 accuracy (Acc@5) of $59.14\%$, and KNN distance of $1.031$. Integrating PAA yields moderate gains, raising Acc@1 to $39.06\%$ and Acc@5 to $67.46\%$, while reducing the KNN distance to $0.975$. In contrast, TAA achieves substantial improvements, boosting Acc@1 to $62.58\%$ and Acc@5 to $84.09\%$, alongside a sharper drop in KNN distance to $0.855$. This suggests that TAA more effectively recovers semantically meaningful gradients that better align with the generator manifold.

Under the stronger NegLS defense, which imposes stronger regularization and suppresses inversion more aggressively, the baseline Acc@1 is just $8.40\%$. Although this setting presents a more challenging scenario, PAA still offers slight improvements, raising Acc@1 to $8.62\%$ and reducing KNN distance from $1.309$ to $1.303$. TAA further improves Acc@1 to $10.61\%$ and reduces KNN distance

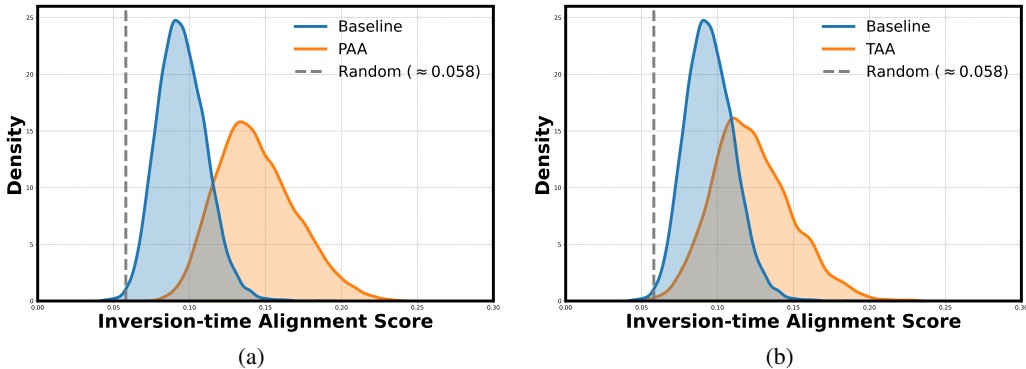

(a)                    (b)

Figure 12: **Distribution of inversion-time alignment scores. (a)** Comparison between baseline and PAA method. **(b)** Comparison between baseline and TAA method. Each plot shows the distribution of alignment scores between the inversion-time loss gradients and the generator manifold. The measurement is performed using the PPA method with a StyleGAN generator trained on FFHQ, and the target model is a ResNet-18 trained on CelebA. Both PAA and TAA lead to a rightward shift in the score distribution, indicating stronger alignment with the generator manifold.

Table 8: Model inversion performance against SOTA defense methods in high-resolution settings. Target model $f$ = ResNet-152, trained on $\mathcal{D}_{\text{pri}}$ = FaceScrub. GAN is pre-trained on $\mathcal{D}_{\text{aux}}$ = FFHQ.

| Method | Acc@1↑ | Acc@5↑ | KNN Dist↓ |
|---|---|---|---|
| No Defense | 57.89 | 81.25 | 0.893 |
| BiDO-HSIC | 35.11 | 59.14 | 1.031 |
| + PAA | 39.06 (+3.95) | 67.46 (+8.32) | 0.975 (-0.056) |
| + TAA | 62.58 (+27.47) | 84.09 (+24.95) | 0.855 (-0.176) |
| NegLS | 8.40 | 23.50 | 1.309 |
| + PAA | 8.62 (+0.22) | 23.67 (+0.17) | 1.303 (-0.006) |
| + TAA | 10.61 (+2.21) | 27.31 (+3.81) | 1.278 (-0.031) |
| TL-DMI | 25.14 | 51.72 | 1.026 |
| + PAA | 34.93 (+9.79) | 63.66 (+11.94) | 1.022 (-0.004) |
| + TAA | 47.80 (+22.66) | 75.51 (+23.79) | 0.971 (-0.055) |

to 1.278. While the absolute gains are smaller due to the strength of the defense, the consistent improvements across all metrics indicate enhanced gradient informativeness.

Finally, the TL-DMI defense, which involves partial model freezing during fine-tuning, the baseline attack achieves Acc@1 of $25.14\%$, Acc@5 of $51.72\%$, and KNN distance of $1.026$. PAA improves Acc@1 to $34.93\%$ and Acc@5 to $63.66\%$, slightly reducing the KNN distance to $1.022$. TAA again shows superior performance, reaching Acc@1 of $47.80\%$, Acc@5 of $75.51\%$, and decreasing KNN distance to $0.971$.

Overall, across all three defenses, both PAA and TAA enhance inversion performance, with TAA consistently outperforming PAA in all metrics. These results highlight the generality and robustness of our alignment-enhancing framework. TAA, in particular, effectively boosts attack success rates while recovering reconstructions that are perceptually and semantically closer to the true data distribution, even under strong privacy-preserving defenses.

### E.3 Ablation Study

In this subsection, we perform an ablation study to examine the sensitivity of our proposed *AlignMI* approach to two key hyperparameters: (1) the number of samples $K$ used to compute the smoothed, alignment-enhanced gradients, and (2) the perturbation strength $\alpha$ used in the perturbation-averaged alignment (PAA) method. All experiments are conducted using a DenseNet-121 target model trained

Table 9: Ablation study on PAA sample size $K$ with $\alpha = 0.03$. Higher $K$ improves results slightly, but gains saturate.

| Method | $K$ | Acc@1↑ | Acc@5↑ | KNN Dist↓ |
|--------|-----|--------|--------|-----------|
| PPA | - | 77.00 | 92.44 | 0.807 |
| + PAA | 20 | 79.56 | 93.24 | 0.804 |
| + PAA | 60 | 78.64 | 92.84 | 0.804 |
| + PAA | 100 | 78.60 | 93.32 | 0.802 |
| + PAA | 150 | 79.16 | 93.44 | 0.797 |

Table 10: Ablation study on PAA sample size $K$ with $\alpha = 0.05$. Higher $K$ improves results slightly, but gains saturate.

| Method | $K$ | Acc@1↑ | Acc@5↑ | KNN Dist↓ |
|--------|-----|--------|--------|-----------|
| PPA | - | 77.77 | 92.73 | 0.798 |
| + PAA | 20 | 82.52 | 94.48 | 0.789 |
| + PAA | 60 | 82.04 | 94.08 | 0.789 |
| + PAA | 100 | 81.92 | 94.55 | 0.788 |
| + PAA | 150 | 82.28 | 94.16 | 0.788 |

Table 11: Ablation study on TAA sample size $K$. Higher $K$ yields marginal gains.

| Method | $K$ | Acc@1↑ | Acc@5↑ | KNN Dist↓ |
|--------|-----|--------|--------|-----------|
| PPA | - | 77.77% | 92.73% | 0.798 |
| + TAA | 20 | 87.64% | 96.04% | 0.748 |
| + TAA | 60 | 88.28% | 96.44% | 0.746 |
| + TAA | 100 | 88.44% | 96.16% | 0.745 |
| + TAA | 150 | 88.16% | 96.44% | 0.745 |

Table 12: Ablation study on PAA perturbation scale $\alpha$ at fixed $K = 60$. Increasing $\alpha$ improves alignment but saturates.

| Method | $\alpha$ | Acc@1↑ | Acc@5↑ | KNN Dist↓ |
|--------|----------|--------|--------|-----------|
| PAA | 0.01 | 74.72% | 91.80% | 0.822 |
| PAA | 0.03 | 78.64% | 92.84% | 0.804 |
| PAA | 0.05 | 82.04% | 94.08% | 0.789 |
| PAA | 0.10 | 82.84% | 94.48% | 0.780 |
| PAA | 0.15 | 79.16% | 93.44% | 0.797 |

on the FaceScrub dataset at $224 \times 224$ resolution, with a StyleGAN generator pre-trained on FFHQ serving as the prior model.

**Effect of Sample Number $K$ in PAA.** We first investigate the influence of the sample number $K$ on PAA under two different perturbation strengths. As shown in Tabs. 9 and 10, we observe that increasing $K$ has a limited effect on attack accuracy, which remains relatively stable across settings. However, the KNN distance continues to decrease slightly as $K$ grows, indicating progressively finer reconstruction fidelity. These findings suggest that while larger $K$ offers marginal improvements, even a relatively small sample number (*e.g.,* $K = 20$) is sufficient to achieve substantial gains over the baseline. This highlights the practicality of PAA in improving inversion performance with minimal computational overhead.

**Effect of Sample Number $K$ in TAA.** We conduct a similar evaluation for the TAA method. As presented in Tab. 11, both attack accuracy and KNN distance improve as $K$ increases, with performance gains tapering off beyond $K = 100$. Notably, TAA achieves strong results even with $K = 20$, outperforming the baseline by a significant margin. This again demonstrates that our training-free alignment promotion strategy enhances inversion performance effectively, even with limited sampling, thus making it computationally efficient.

**Effect of Perturbation Strength $\alpha$ in PAA.** Finally, we analyze the role of the perturbation strength $\alpha$ in PAA. As shown in Tab. 12, increasing $\alpha$ initially boosts both attack accuracy and KNN distance, with performance peaking around $\alpha = 0.1$. However, beyond this threshold (*e.g.,* $\alpha = 0.15$), both metrics begin to deteriorate, likely due to the perturbations introducing excessive noise that destabilizes the model's prediction and results in unreliable gradients. This suggests that careful tuning of $\alpha$ is critical, and moderate values around $0.05$ to $0.1$ provide a favorable balance between denoising and preserving informative signals.

### E.4 Visualization of Gradient Images

In this subsection, we qualitatively demonstrate that both PAA and TAA produce loss gradients that are better aligned with the generator manifold. Our analysis focuses on the high-resolution setting, which enables high-quality visualizations of gradient structures. Figs. 13, 14, and 15 present gradient visualizations from ResNet-18, DenseNet-121, and ResNeSt-50 models trained on CelebA. Each figure compares gradient maps produced by the baseline, PAA, and TAA methods, using GANs pre-trained on FFHQ. We also visualize the inversion-time loss gradient images for three attack methods in the low-resolution setting (see Fig. 16), as a complementary comparison to Fig. 1(b).

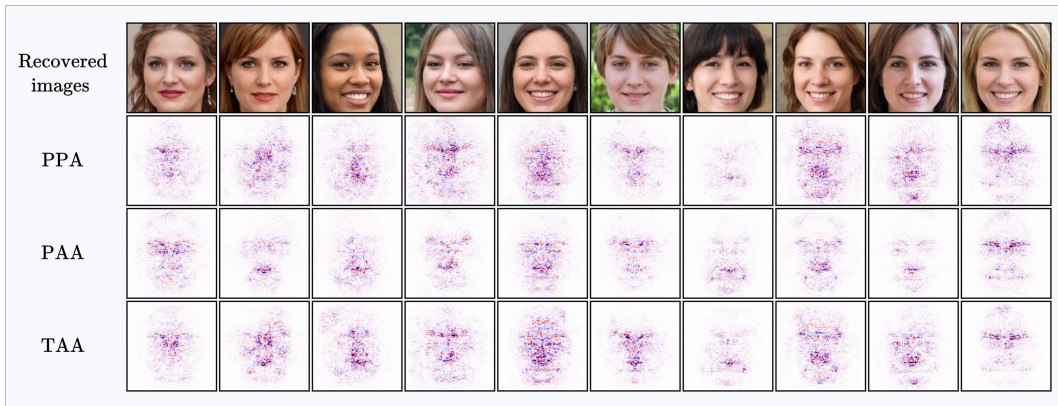

Figure 13: Visual comparison of inversion-time loss gradients for PPA in the high-resolution setting. We illustrate reconstructed samples for ten classes in $\mathcal{D}_{\mathrm{pri}}$ = CelebA using GANs pre-trained on $\mathcal{D}_{\mathrm{aux}}$ = FFHQ. The target model is ResNet-18. (Best viewed with zoom.)

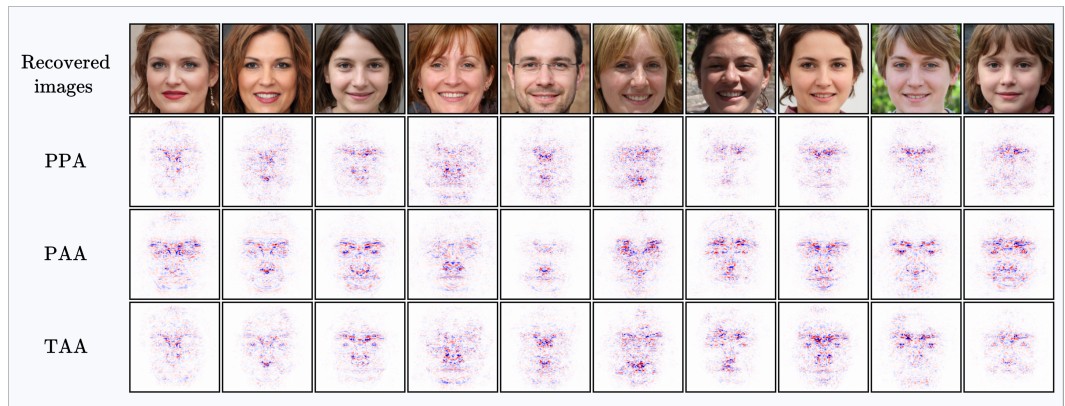

Figure 14: Visual comparison of inversion-time loss gradients for PPA in the high-resolution setting. We illustrate reconstructed samples for ten classes in $\mathcal{D}_{\mathrm{pri}}$ = CelebA using GANs pre-trained on $\mathcal{D}_{\mathrm{aux}}$ = FFHQ. The target model is DenseNet-121. (Best viewed with zoom.)

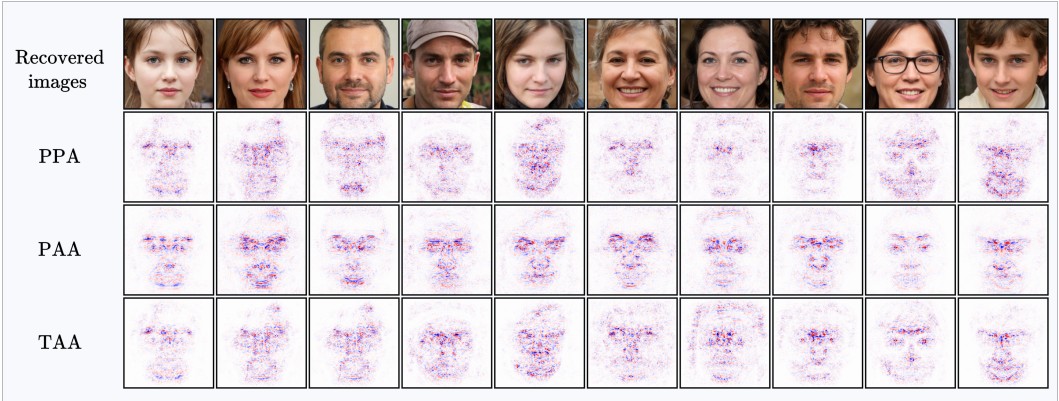

Figure 15: Visual comparison of inversion-time loss gradients for PPA in the high-resolution setting. We illustrate reconstructed samples for ten classes in $\mathcal{D}_{\mathrm{pri}}$ = CelebA using GANs pre-trained on $\mathcal{D}_{\mathrm{aux}}$ = FFHQ. The target model is ResNeSt-50. (Best viewed with zoom.)

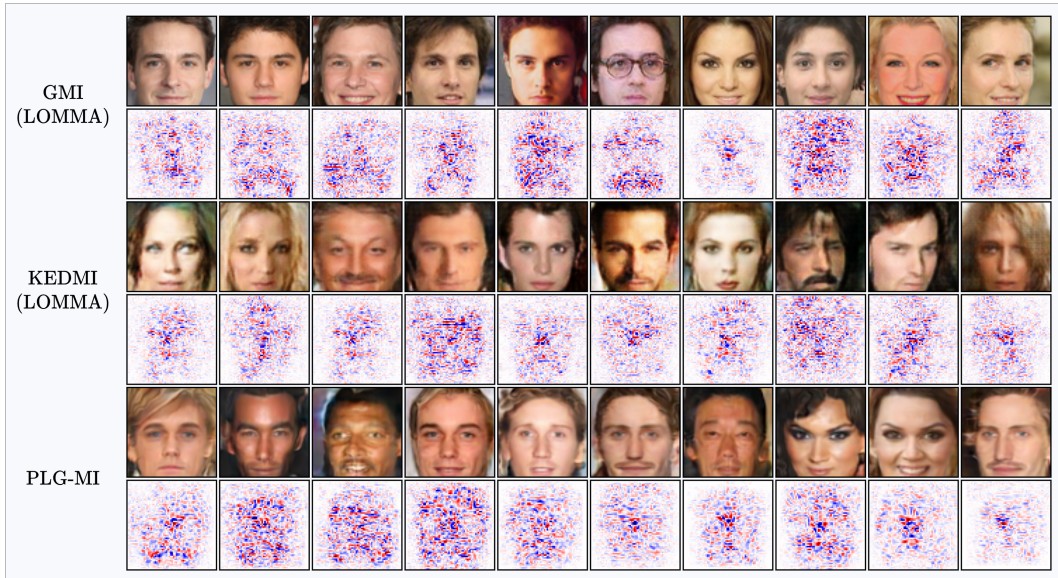

Figure 16: Visual of inversion-time loss gradients for three attack methods in the low-resolution setting. The target model is FaceNet. (Best viewed with zoom.)

## E.5 Visualization of Reconstructed Images

In this subsection, we present qualitative results of the baseline attack methods and our proposed AlignMI approach. High-resolution reconstructions are shown in Figs. 17 and 18. Fig. 17 compares reconstructed samples from the first ten classes using ResNet-18, DenseNet-121, and ResNeSt-50 trained on CelebA, with GANs pre-trained on FFHQ. Fig. 18 provides similar results for the same target models trained on FaceScrub, also using FFHQ-pretrained GANs.

In low-resolution setting, we evaluate reconstruction quality by comparing samples from the first ten classes generated by GMI (LOMMA) and KEDMI (LOMMA) attack methods. These experiments employ VGG16 and FaceNet trained on CelebA as target models, with GANs pre-trained on both CelebA and FFHQ datasets, as shown in Figs. 19, and 20 respectively. Additionally, we present PLG-MI reconstructions on FaceNet using GANs trained on FFHQ and FaceScrub datasets in Fig. 21.

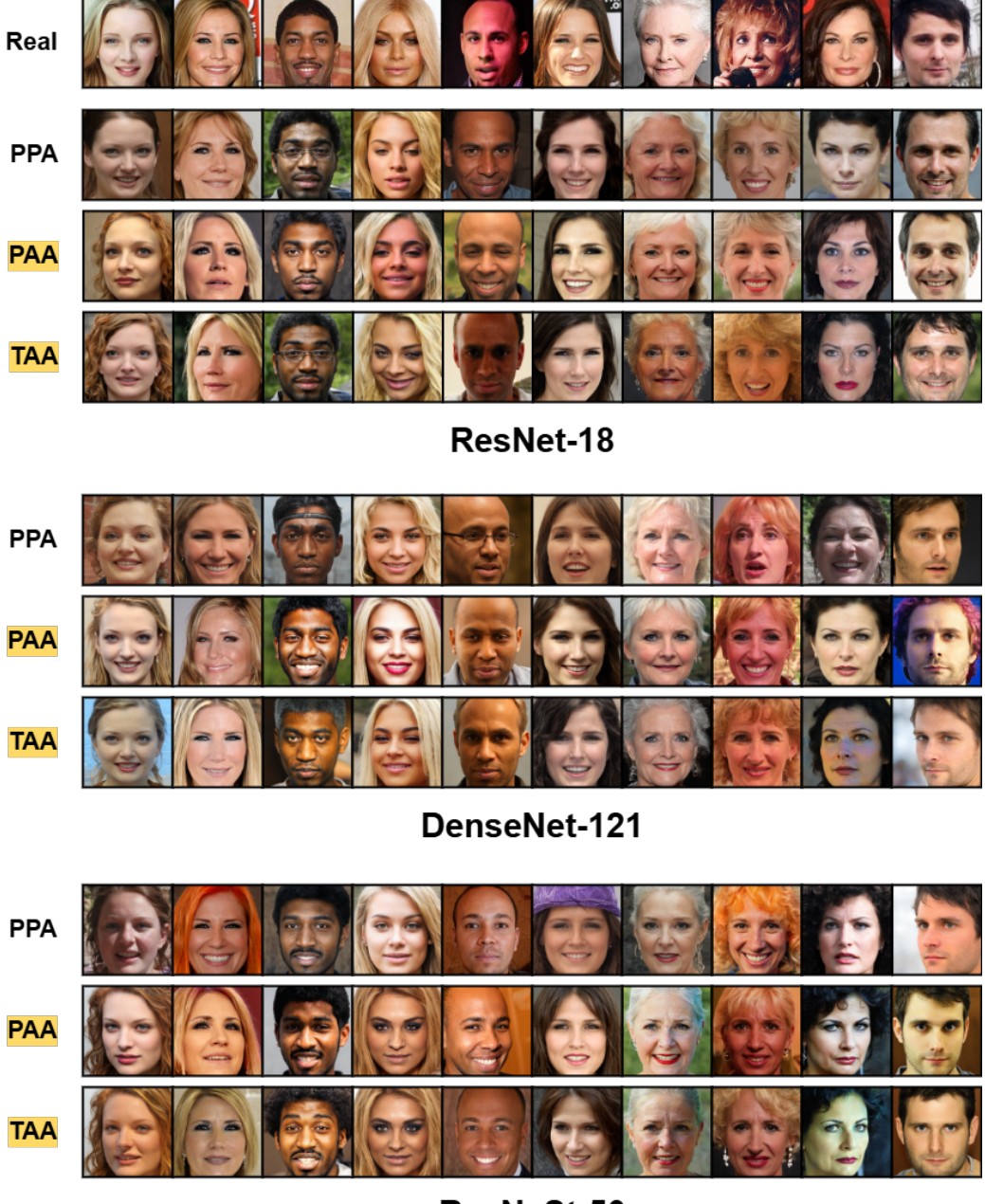

Figure 17: Visual comparison in high-resolution settings. We illustrate reconstructed samples for the first ten classes in $\mathcal{D}_{\mathrm{pri}}$ = CelebA using GANs pre-trained on $\mathcal{D}_{\mathrm{aux}}$ = FFHQ.

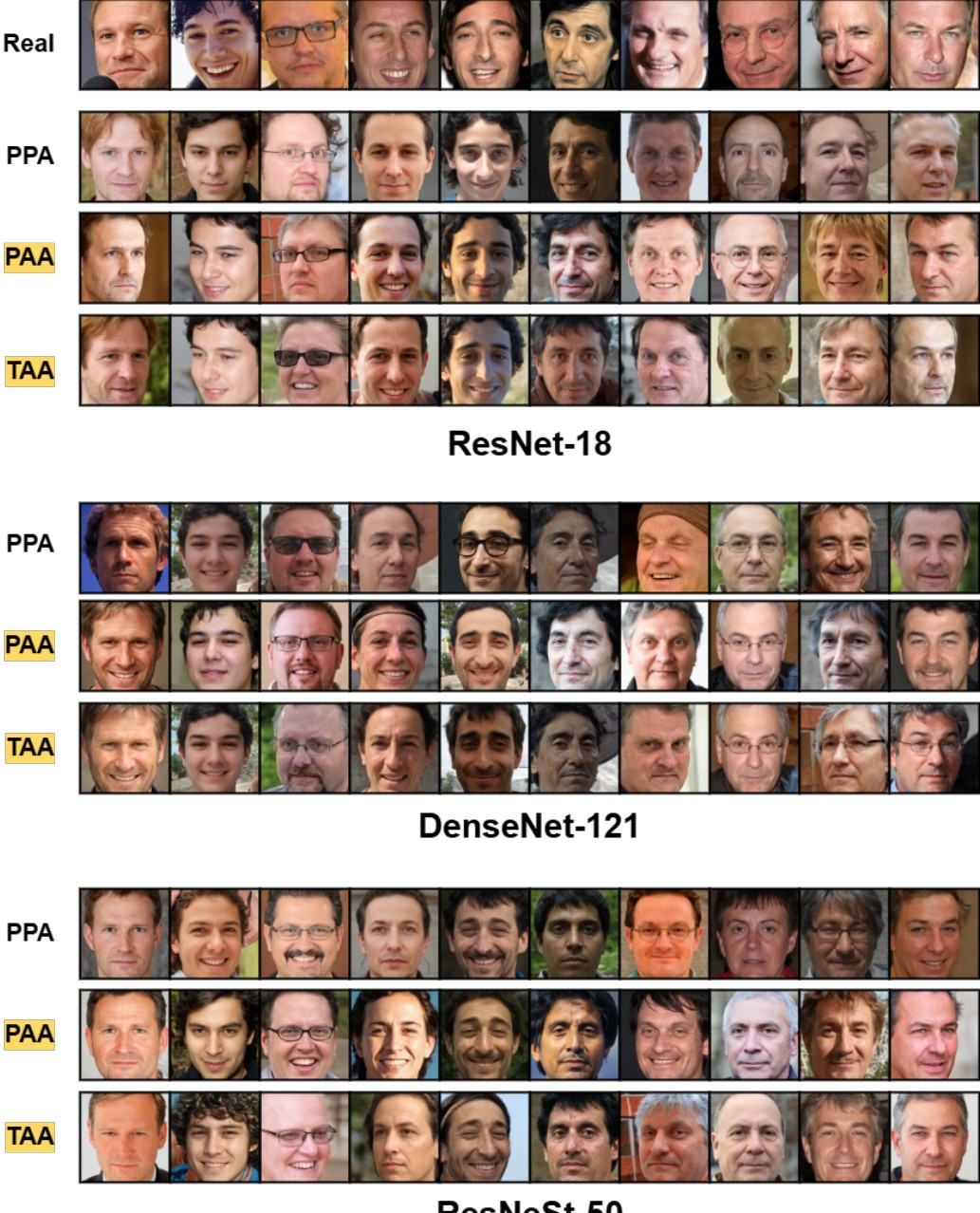

Figure 18: Visual comparison in high-resolution settings. We illustrate reconstructed samples for the first ten classes in $\mathcal{D}_{pri}$ = FaceScrub using GANs pre-trained on $\mathcal{D}_{aux}$ = FFHQ.

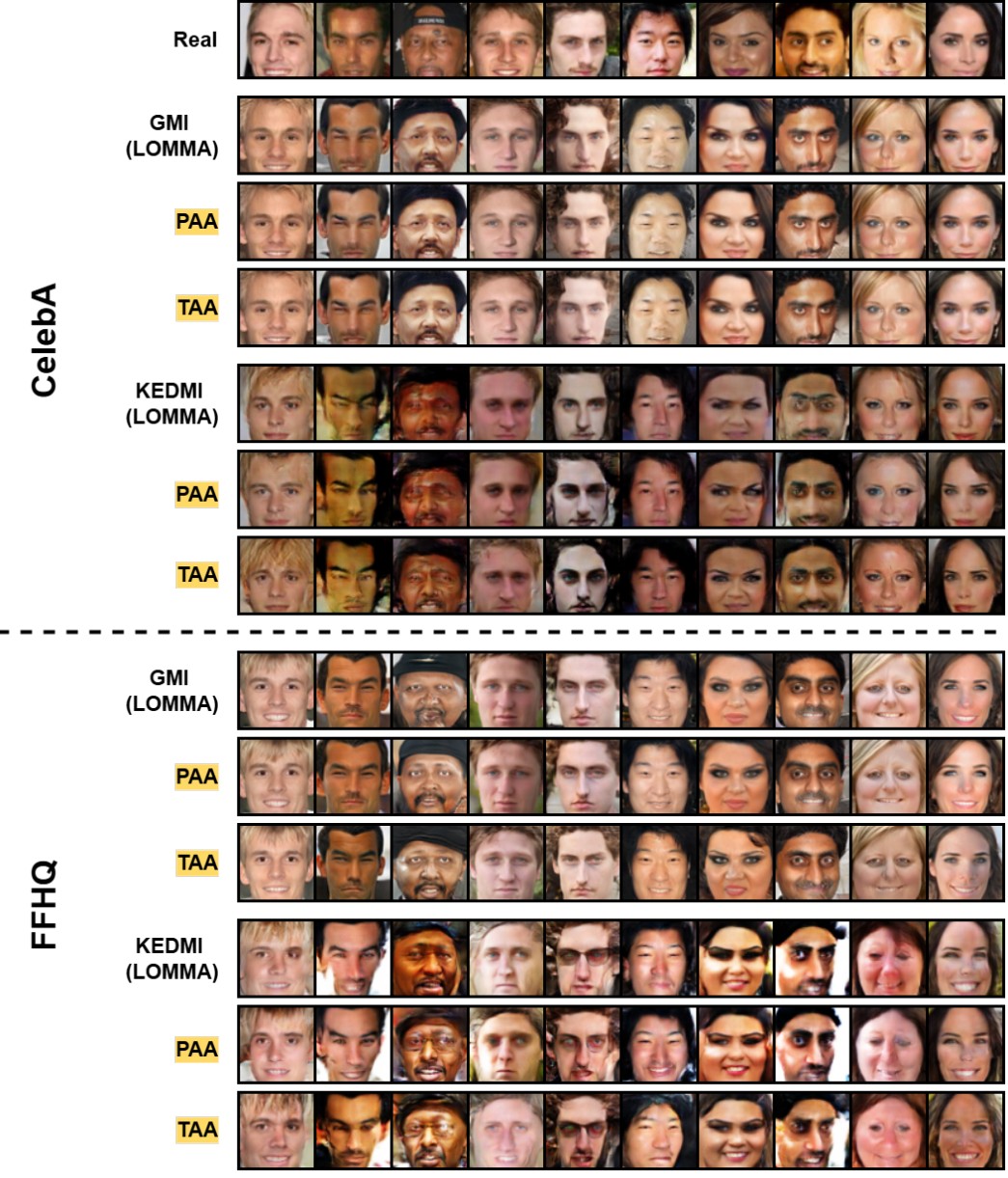

Figure 19: Visual comparison in low-resolutions settings. We illustrate reconstructed samples for the first ten classes in $\mathcal{D}_{pri}$ = CelebA using GANs trained from scratch on $\mathcal{D}_{aux}$ = CelebA / FFHQ. The target model is VGG16.

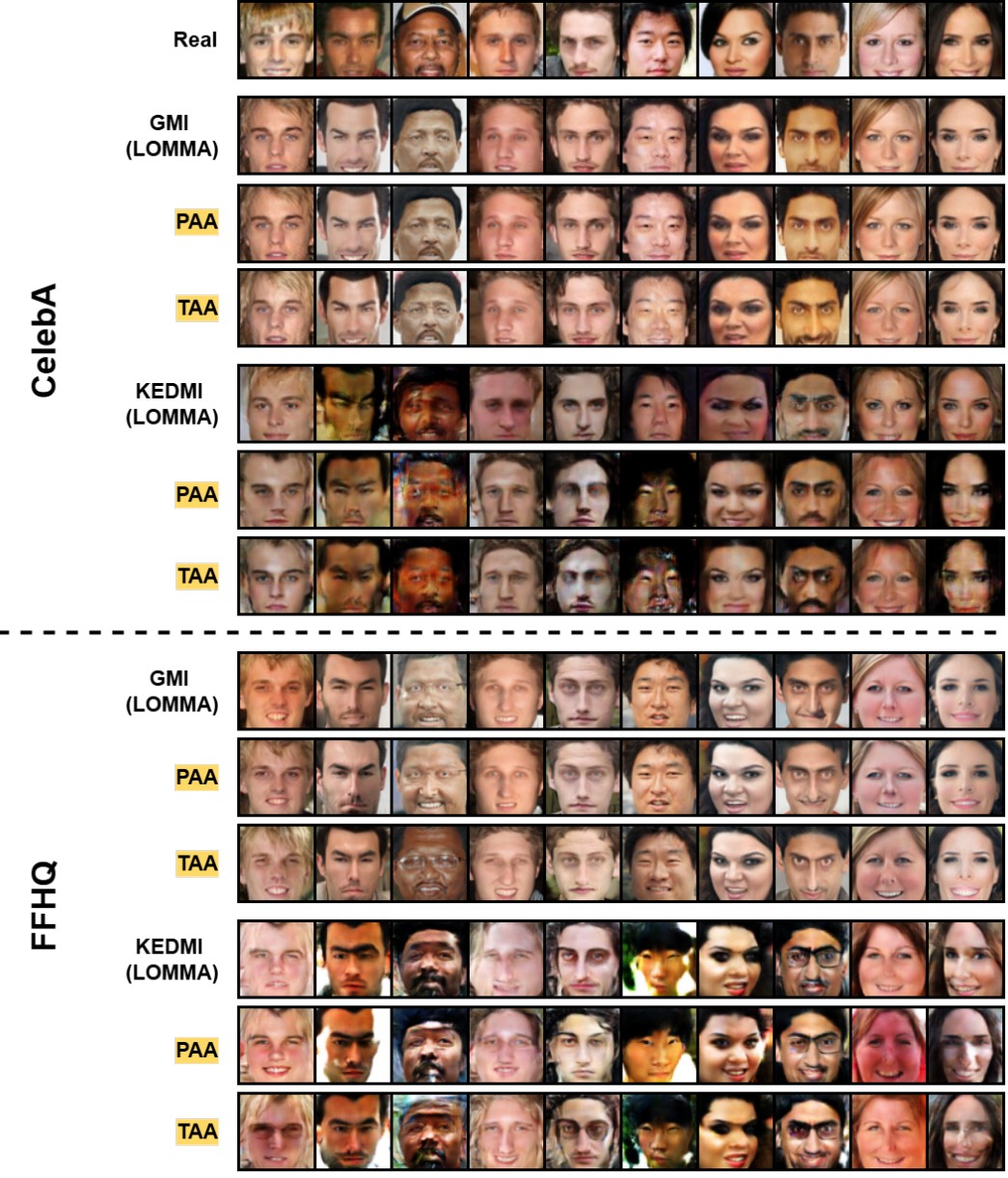

Figure 20: Visual comparison in low-resolutions settings. We illustrate reconstructed samples for the first ten classes in $\mathcal{D}_{pri}$ = CelebA using GANs trained from scratch on $\mathcal{D}_{aux}$ = CelebA / FFHQ. The target model is FaceNet.

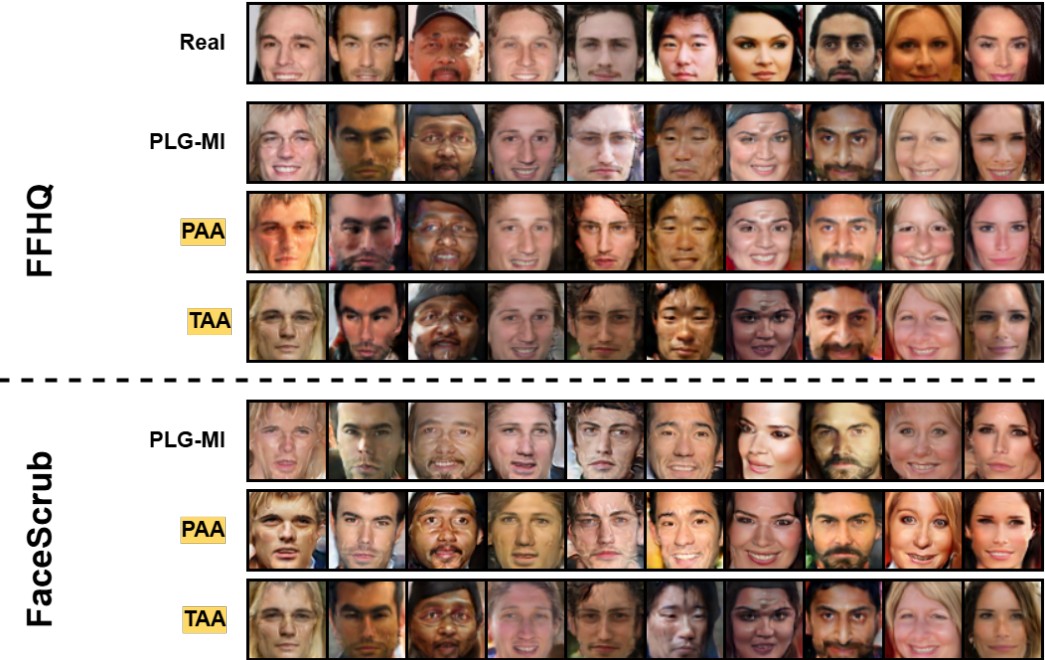

Figure 21: Visual comparison in low-resolutions settings. We illustrate reconstructed samples for the first ten classes in $\mathcal{D}_{\text{pri}}$ = CelebA using GANs trained from scratch on $\mathcal{D}_{\text{aux}}$ = FFHQ / FaceScrub. The target model is FaceNet.

