# OpenReview forum: "Generative Model Inversion Through the Lens of the Manifold Hypothesis"
_NeurIPS.cc/2025/Conference — NeurIPS 2025 poster_

### Official Review · Reviewer_6LmF · 2025-06-26

**Clarity:** 2
**Significance:** 2
**Originality:** 3
**Rating:** 4
**Confidence:** 4

**Summary:**

This manuscript investigates generative model inversion attacks (MIAs), which aim to reconstruct class-representative samples from trained models. The authors analyze the mechanism of generative MIAs through a geometric lens, revealing that these attacks implicitly denoise inversion loss gradients by projecting them onto the tangent space of the generator manifold, thus preserving informative directions aligned with the data manifold. They hypothesize that models are more vulnerable to MIAs when their loss gradients align more closely with the generator manifold, validated by designing an alignment-aware training objective. Additionally, they introduce AlignMI, a training-free approach that enhances gradient-manifold alignment via perturbation-averaged (PAA) and transformation-averaged (TAA) strategies, outperforming state-of-the-art generative MIAs in experiments. Notable limitations include statistical imprecision in experiments, increased computational overhead, and potential discrepancies between estimated and real data manifolds.

**Questions:**

- The author's research subject in this manuscript is generative MIA. What is the difference between this and model inversion (such as psp, e4e, etc.)? Where is the specific manifestation of the attack? I hope the author can clarify these conceptual differences.

- The datasets targeted by the author are all facial datasets, and most of them are cropped images. As is well known, the degree of restoration of the background area in image inversion is an indicator of the quality of inversion. However, I did not find any significant information from the results presented in this manuscript.

- The manuscript did not report experimental error bars or confidence intervals, and only generated at least 100 samples through a single attack to reduce randomness. This approach cannot fully validate the reliability of the results, especially as there may be fluctuations in generalization across different models and datasets.

- Only a few SOTA defenses such as BiDO and NegLS were compared, without covering emerging defense strategies such as adversarial training and gradient masking, which may not fully reflect the robustness of the method.

**Ethical Concerns:**

["NO or VERY MINOR ethics concerns only"]

**Final Justification:**

The authors should ensure consistency in hyperparameter settings across experiments to maintain fairness. However, this issue does not undermine the potential impact of the work in its field. Based on all the above comments and the authors’ responses, and from my experience, the intention to accept this paper outweighs the intention to reject it.

**Limitations:**

- The author claims that the method proposed in this manuscript is aimed at generative models. Generative models include generative flow, VAE, GAN, and diffusion models. Is the method of this manuscript applicable to all generative models?

- Assuming that the attacker has white box access to the model (such as gradient information), but in black box scenarios (where only the model output can be observed), the effectiveness of methods such as AlignMI may be significantly reduced.

- Generative MIA highly relies on the quality of pre trained GANs (such as StyleGAN). If the generator cannot capture key features of private data (such as specific poses and expressions), the inversion effect will be significantly reduced.

- The projection operation in aligning training targets has extremely high computational costs in high-dimensional inputs or multi class scenes. Although upper bound optimization reduces overhead, it may still limit its application in large-scale tasks.

**Quality:**

2

**Strengths And Weaknesses:**

**Strengths:**

- The proposed training objective promotes input gradient alignment with the data manifold (estimated via a pre-trained VAE), enhancing loss gradient informativeness during inversion. This approach validates that stronger alignment increases model vulnerability.

- By averaging gradients from perturbed or transformed inputs, AlignMI suppresses off-manifold noise, yielding consistent improvements.

**Weaknesses:**

Please see the details in Questions and Limitations.

---

> ### Author Rebuttal · Authors · 2025-07-31
>
> Thank you for your great comments and efforts in reviewing our paper. Please see our detailed responses to your comments and suggestions below.
>
>
> > Q1. What is the difference between this and model inversion (such as psp, e4e, etc.)? Where is the specific manifestation of the attack?
>
>
> Thank you for your question. To clarify the distinction between **generative model inversion Attacks (MIA)** and model inversion methods such as **PSP** and **e4e**, we can identify several key differences in both their approaches and objectives.
>
> **MIA** is a category of **privacy attack** aimed at reconstructing representative samples by exploiting access to **only a well-trained model**. Early work [r1] framed MIAs as an input-space optimization problem, where the objective is to use gradient descent to find inputs that maximize the prediction score for a given target class.
>
> **Generative MIA** [r2] extends this framework by integrating generative models such as GANs to learn an image prior. This prior is used to constrain the inversion process to the latent space of the generator, significantly improving the visual quality and semantic relevance of the reconstructed samples, which enhances the effectiveness of model inversion attacks.
>
> In contrast, **model inversion methods** like PSP and E4E focus on inverting a pre-trained generative model (e.g., StyleGAN) to generate latent representations or embeddings for **a given target image**. This enables tasks such as image manipulation and editing, where the goal is not necessarily to infer sensitive information but to alter or reconstruct the image from the latent representation.
>
> We hope this clarifies the conceptual differences between these two approaches. Please let us know if further clarification is needed!
>
> > Q2. The datasets targeted by the author are all facial datasets ... in this manuscript.
>
>
> Thank you for your question. We followed the common evaluation protocol established in prior MIA literature [r2-4], which predominantly uses facial datasets due to their relevance to real-world privacy concerns. These datasets reflect scenarios where the primary risk lies in revealing sensitive attributes such as facial features. As we clarified earlier, the goal of MIA is to reconstruct representative samples that reveal class-sensitive information. Consequently, the background area does not carry significant class-sensitive information, and therefore is less critical to the effectiveness of the inversion in our context.
>
> > Q3. The manuscript did not report experimental error bars or confidence intervals.
>
>
> Thank you for pointing this out. To address this concern, we have conducted three independent runs (with different random seeds) of our key experiments in Table 2 to assess the consistency and statistical significance of our results. The averaged results and standard deviations are reported in the following table:
>
>
> |              |        | **CelebA**        |                   | **FaceScrub**     |                   |
> |--------------|--------|-------------------|-------------------|-------------------|-------------------|
> | **Target Model** | Method | Acc@1 $\uparrow$  | KNN Dist $\downarrow$ | Acc@1 $\uparrow$  | KNN Dist $\downarrow$ |
> | **ResNet‑18**    | PPA    | 86.04 $\pm$ 0.04  | 0.6897 $\pm$ 0.0002 | 81.86 $\pm$ 0.35  | 0.7943 $\pm$ 0.0024 |
> |              | + PAA (ours) | 88.69 $\pm$ 0.28  | 0.6691 $\pm$ 0.0011 | 83.98 $\pm$ 0.22  | 0.7761 $\pm$ 0.0025 |
> |              | + TAA (ours) | 90.17 $\pm$ 1.15  | 0.6624 $\pm$ 0.0001 | 93.46 $\pm$ 0.30  | 0.6984 $\pm$ 0.0071 |
> | **DenseNet‑121** | PPA    | 81.15 $\pm$ 0.80  | 0.7148 $\pm$ 0.0054 | 74.52 $\pm$ 1.78  | 0.8007 $\pm$ 0.0178 |
> |              | + PAA (ours) | 85.42 $\pm$ 0.23  | 0.6870 $\pm$ 0.0005 | 80.35 $\pm$ 0.06  | 0.7363 $\pm$ 0.0021 |
> |              | + TAA (ours) | 87.84 $\pm$ 0.73  | 0.6832 $\pm$ 0.0095 | 84.91 $\pm$ 0.14  | 0.7287 $\pm$ 0.0036 |
> | **ResNeSt‑50**   | PPA    | 71.16 $\pm$ 0.10  | 0.7926 $\pm$ 0.0004 | 71.62 $\pm$ 0.20  | 0.8315 $\pm$ 0.0002 |
> |              | + PAA (ours) | 76.08 $\pm$ 0.17  | 0.7614 $\pm$ 0.0012 | 73.52 $\pm$ 0.55  | 0.8074 $\pm$ 0.0042 |
> |              | + TAA (ours) | 78.07 $\pm$ 1.41  | 0.7665 $\pm$ 0.0120 | 84.10 $\pm$ 0.04 | 0.7568 $\pm$ 0.0005 |
>
>
>
> These results confirm that our method consistently outperforms the baselines across runs, demonstrating statistical reliability. We will include the updated results in the main results section (Section 6.3) of the final version of our paper to enhance the reliability of the proposed method.
>
>
>
> > Q4. Only a few SOTA defenses such as BiDO and NegLS were compared ... the robustness of the method.
>
>
> Thank you for your comment. We chose to evaluate with BiDO, NegLS, and TL-DMI because **they represent state-of-the-art defense mechanisms specifically designed to counter model inversion attacks**. However, as we understand it, adversarial training and gradient masking are primarily techniques designed to defend against adversarial attacks (please correct us if our understanding is incorrect), rather than directly addressing the unique vulnerabilities posed by MIAs. As a result, these methods fall outside the scope of our evaluation.
>
>
> > L1. Is the method of this manuscript applicable to all generative models?
>
>
> Thank you for your question. The method proposed in this manuscript is applicable to **single-step generative models** such as generative flow, VAE, and GAN, where the generation process can be expressed as $\mathbf{x}=G(\mathbf{z})$ through a single forward pass of the generator. This structure enables direct computation of the Jacobian, allowing us to apply projection-based methods for analysis.
>
> However, for **diffusion models**, the sampling/generation process is iterative and involves denoising at each step, which makes it more complex and fundamentally different from single-step models. As a result, our projection-based method is not directly applicable to diffusion models.
>
>
> > L2. Re: Extending AlignMI to black-box settings.
>
>
> Thank you for your comment. As stated in our problem setup (Section 2), our study focuses on the **white-box setting**, since AlignMI relies on inversion loss gradients to improve gradient–manifold alignment—this naturally requires access to the model's internal gradients.
>
> We acknowledge that model inversion attacks have also been studied in black-box settings, where several works have proposed search-based methods in the latent space to approximate gradients using surrogate signals [r5-6]. While our method is not directly applicable in such scenarios, we believe **the core idea of promoting alignment between surrogate gradients and the data manifold** holds potential for generalization to the black-box setting. We view this as a promising direction for future research.
>
> > L3. Generative MIA highly relies on the quality of pre-trained GANs (such as StyleGAN) ... the inversion effect will be significantly reduced.
>
> Thank you for your comment. The goal of MIA is to reconstruct representative samples that reveal class-sensitive information, and therefore specific poses and expressions may not be critical to our context. We acknowledge that the distributional gap between the generator and the private data can affect inversion quality [r7]. However, recent studies have shown that characteristic features of target classes can still be revealed even under large distributional shifts [r3]. Moreover, while this limitation exists, it primarily arises from the choice and quality of the generator itself. Addressing this issue **falls outside the scope of our work**, which focuses on enhancing inversion through gradient-manifold alignment.
>
> > L4. The projection operation in aligning ... it may still limit its application in large-scale tasks.
>
> Thank you for your comment. Compared to the standard classification objective, the primary computational overhead of the gradient-manifold alignment training objective comes from the additional **single matrix-vector multiplication** required for the projection operation. However, this increase in computational cost is marginal relative to the overall time spent on training.
>
> To further investigate this, we tested the gradient-manifold alignment training objective against the standard classification objective across three models. We conducted experiments over 30 epochs, with each experiment repeated three times. The results are as follows:
>
>
> | Target Model | $L_\text{CE}$    | $L_\text{Align}$    | Relative Increase |
> |:------------:|:------------------------:|:----------------------------:|:-------------------:|
> | **VGG16**    | 442.21 $\pm$ 5.18      | 449.04 $\pm$ 2.20          | 1.54 %              |
> | **FaceNet64**| 438.18 $\pm$ 3.39      | 444.53 $\pm$ 2.98         | 1.45 %              |
> | **IR152**    | 436.84 $\pm$ 4.20      | 455.99 $\pm$ 5.09         | 4.38 %              |
>
> These findings suggest that while there is a slight increase in computational cost, it has **negligible impact** on overall training efficiency.
>
> **We hope our responses have clarified the confusion and addressed your concerns. We would greatly appreciate it if you could take them into consideration during your final evaluation of our work. Please let us know if you have any outstanding questions.**
>
>
>
> ---
> **References:**
>
> [r1] Fredrikson et al. Model inversion attacks that exploit confidence information and basic countermeasures. In CCS, 2015.
>
> [r2] Zhang et al. The secret revealer: Generative model-inversion attacks against deep neural networks. In CVPR, 2020.
>
> [r3] Struppek et al. Plug & play attacks: Towards robust and flexible model inversion attacks. In ICML, 2022.
>
> [r4] Peng et al. Pseudo-private data guided model inversion attacks. In NeurIPS, 2024.
>
> [r5] An et al. Mirror: Model inversion for deep learning network with high fidelity. In NDSS, 2022.
>
> [r6] Kahla et al. Label-only model inversion attacks via boundary repulsion. In CVPR, 2022.

---

> > ### Comment · Reviewer_6LmF · 2025-08-01
> > **Response to Rebuttal of Authors**
> >
> > I have read the author's response, which has addressed most of my questions. However, the following points still require clarification from the author to support my rating:
> >
> > - Why are some of the error bars in the response to Q3 (ResNet-18 + TAA, ResNet-50 + TAA) so large? The author needs to provide an explanation for this. In addition to accuracy, the robustness of the model is also a relatively important indicator. Furthermore, the author needs to supplement more experiments to support this issue (the author is not required to do so during the author-reviewer rebuttal period).
> >
> > - Why is the KNN Dist value in Table 1 of the original text so high, while the values in other tables are relatively low?
> >
> > - The author needs to explain the meanings of the two indicators, $L_{CE}$ and $L_{Align}$, in L4.

---

> ### Author Response · Authors · 2025-08-03
> **Response to Reviewer 6LmF’s Remaining Concerns**
>
> We are glad to have resolved most of your concerns and would now like to address the remaining points with further clarification.
>
> > Q1. Why are some of the error bars in the response to Q3 (ResNet-18 + TAA, ResNeSt-50 + TAA) so large?
>
> Thank you for your question. Due to time constraints during the rebuttal period, we initially conducted three independent runs per setting. This limited number of trials may have contributed to slightly higher standard deviations for certain configurations (i.e., ResNet-18 + TAA, ResNeSt-50 + TAA), as you point out:
>
>
> | **Target Model** | Method | Acc@1 $\uparrow$  | KNN Dist $\downarrow$ |
> |------------------|--------|------------------|-----------------------|
> | **ResNet‑18**    | PPA    | 86.04 $\pm$ 0.04  | 0.6897 $\pm$ 0.0002    |
> |                  | + TAA (ours)  | 90.17 $\pm$ 1.15  | 0.6624 $\pm$ 0.0001    |
> | **ResNeSt‑50**   | PPA    | 71.16 $\pm$ 0.10  | 0.7926 $\pm$ 0.0004    |
> |                  | + TAA (ours) | 78.07 $\pm$ 1.41  | 0.7665 $\pm$ 0.0120    |
>
> While these values may appear relatively large compared to other entries, we would like to emphasize that **the absolute magnitudes of the deviations remain modest**. More importantly, **even within these variance bounds, the TAA configurations consistently outperform the baseline (PPA) across all evaluated setups.**
>
> To further address this concern, we have added 5 more runs for both cases. The updated results are as follows:
>
> | **Target Model** | Method        | Acc@1 $\uparrow$  | KNN Dist $\downarrow$ |
> |------------------|---------------|------------------|-----------------------|
> | **ResNet‑18**    | PPA           | 86.04 $\pm$ 0.04  | 0.6897 $\pm$ 0.0002    |
> |                  | + TAA (ours)  | 89.94 $\pm$ 0.10  | 0.6671 $\pm$ 0.0006    |
> | **ResNeSt‑50**   | PPA           | 71.16 $\pm$ 0.10  | 0.7926 $\pm$ 0.0004    |
> |                  | + TAA (ours)  | 78.13 $\pm$ 0.16  | 0.7642 $\pm$ 0.0002    |
>
> These additional experiments show that the initially observed variance diminishes with increased repetitions, confirming the robustness and consistency of our proposed method.
>
>
>
> > Q2. Why is the KNN Dist value in Table 1 of the original text so high, while the values in other tables are relatively low?
>
>
> Thank you for your observation. The discrepancy in KNN distance values between Table 1 and Table 2 in the manuscript primarily arises from **differences in the feature extractors used across resolution settings**. In Table 1 (hypothesis validation), we adopt a **low-dimensional setting (64×64)**. as computing and storing the decoder Jacobian at high resolution (224×224) is prohibitively expensive. Since the goal here is to validate the main hypothesis rather than implement a practical attack, experiments in low-dimensional settings are sufficient and appropriate for empirical analysis. For the main experiments in Table 2, we evaluate the effectiveness of AlignMI under **high-resolution settings (224×224)**, which reflect more realistic model inversion attack scenarios.
>
> As for the KNN distance metric: It measures the $\ell_2$ distance between the reconstructed image and its nearest neighbor in the training set, computed in an embedding space. **Following established MIA literature [r1–r2], we use different feature extractors for different resolutions**. In high-resolution settings, we employ the penultimate layer of InceptionResnetV1 (224×224), which outputs **normalized feature vectors**, resulting in smaller KNN distance values. In low-resolution settings, we use the penultimate layer of FaceNet (64×64), which **does not normalize its feature embeddings**, thereby yielding larger raw distances. Thus, the observed difference in KNN distance values reflects inherent differences in feature extractor behavior rather than any inconsistency in our evaluation methodology.
>
>
>
> > Q3. The author needs to explain the meanings of the two indicators, $L_\text{CE}$ and $L_\text{Align}$ in L4.
>
>
> Thank you for your question. We are happy to clarify the meanings of the two indicators $L_\text{CE}$ and $L_\text{Align}$ mentioned in L4. (1) $L_\text{CE}$ refers to the **standard cross-entropy (CE) loss**, which is the conventional objective used for training classification models. (2) $L_\text{Align}$ refers to the **total loss used in alignment-aware training**, as defined in **Equation (7)** of our paper. This objective augments the cross-entropy loss with an **alignment promotion term**, which encourages the model’s input gradients to align with the estimated tangent space of the data manifold.
>
> In the timing experiments (L4), these two terms are used to compare the training efficiency between standard training and alignment-aware training.
>
>
> ---
> **References:**
>
> [r1] Zhang et al. The secret revealer: Generative model-inversion attacks against deep neural networks. In CVPR, 2020.
>
> [r2] Struppek et al. Plug & play attacks: Towards robust and flexible model inversion attacks. In ICML, 2022.

---

> > ### Comment · Reviewer_6LmF · 2025-08-03
> > **Response to *Response to Reviewer 6LmF’s Remaining Concerns***
> >
> > Thank you for the timely response. To further enhance my understanding of the manuscript, I would appreciate it if the authors could clarify the following two points:
> >
> > - Regarding the Response to Q1, the authors claim that the difference in error bars is due to the number of repeated experiments. I am particularly curious how increasing the number of runs by just two could reduce the error bar from 1.41 to 0.16 (for ResNet-50+TAA). The authors are expected to provide the original data to support this explanation.
> >
> > - Is the magnitude of the KNN distance explicitly related to the dimensionality?

---

> > > ### Author Response · Authors · 2025-08-04
> > > **Response to Reviewer 6LmF’s Remaining Concerns**
> > >
> > > > Q1. Regarding the Response to Q1, the authors claim that the difference in error bars is due to the number of repeated experiments. I am particularly curious how increasing the number of runs by just two could reduce the error bar from 1.41 to 0.16 (for ResNet-50+TAA). The authors are expected to provide the original data to support this explanation.
> > >
> > > Thank you for your follow-up question. Below, we provide the original data for ResNeSt-50 + TAA, which includes both the initial 3 runs and the additional 5 runs:
> > >
> > >
> > > | Setting      | Method           | Acc@1 $\uparrow$       | KNN Dist $\downarrow$      |
> > > |--------------|------------------|----------------|------------------|
> > > | **Initial 3 runs** | Run 1 ($K$=50)     | 79.46          | 0.7555           |
> > > |              | Run 2 ($K$=35)     | 78.10          | 0.7643           |
> > > |              | Run 3 ($K$=20)     | 76.64          | 0.7793           |
> > > | **Additional 5 runs**   | Run 1 ($K$=35)     | 78.26          | 0.7639           |
> > > |              | Run 2             | 78.19          | 0.7642           |
> > > |              | Run 3             | 78.17          | 0.7640           |
> > > |              | Run 4             | 78.01          | 0.7644           |
> > > |              | Run 5             | 77.90          | 0.7644           |
> > >
> > > To ensure a stable evaluation of our method’s robustness, we varied both the random seed and the hyperparameter $K$, which denotes the number of samples used to approximate the expectation in the loss gradient term defined in Equation (8). Specifically, we experimented with $K$ = 50, 35, and 20.
> > >
> > > While this variation had negligible impact in most configurations, it led to slightly larger variance in some setups, e.g., ResNeSt-50 + TAA (78.07 $\pm$ 1.41) when $K$ was varied across runs. To further investigate, we conducted five additional runs with $K$ fixed at 35. The results showed substantially reduced variance and improved stability, suggesting that using a moderately larger and fixed $K$ leads to more reliable performance.
> > >
> > > If you have a preference between reporting results with a fixed or varying $K$, we would be glad to align with it in the final version.
> > >
> > >
> > > > Is the magnitude of the KNN distance explicitly related to the dimensionality?
> > >
> > > A: Thank you for the question. The magnitude of the KNN distance is determined by the feature extractor used to compute the embedding space in which the distance is measured. As previously clarified, In high-resolution settings, we employ the penultimate layer of InceptionResnetV1 (224×224), which outputs **normalized feature vectors**, resulting in smaller KNN distance values. In low-resolution settings, we use the penultimate layer of FaceNet (64×64), which **does not normalize its feature embeddings**, thereby yielding larger raw distances.

---

> > > > ### Comment · Reviewer_6LmF · 2025-08-04
> > > > **Response to *Response to Reviewer 6LmF’s Remaining Concerns***
> > > >
> > > > The authors should ensure consistency in hyperparameter settings across experiments to maintain fairness. However, this issue does not undermine the potential impact of the work in its field. Based on all the above comments and the authors’ responses, and from my experience, the intention to accept this paper outweighs the intention to reject it. Thanks.

---

> > > > > ### Author Response · Authors · 2025-08-04
> > > > >
> > > > > Dear Reviewer 6LmF,
> > > > >
> > > > > Thank you for your thoughtful feedback and positive assessment. We will ensure consistency across all experimental settings and conduct multiple runs to strengthen the robustness of our results in the final version of our paper. Detailed descriptions will be included in the implementation section to enhance the clarity and reproducibility of the paper.
> > > > >
> > > > > Sincerely,
> > > > >
> > > > > The Anonymous Authors

---

### Official Review · Reviewer_WVQ7 · 2025-07-01

**Clarity:** 3
**Significance:** 3
**Originality:** 3
**Rating:** 5
**Confidence:** 3

**Summary:**

This paper presents an in-depth analysis of Model inversion attacks (MIAs), revealing that the input loss gradients involved in MIAs are often noisy and significantly misaligned with the underlying data manifold. The authors empirically demonstrate that when a model’s loss gradients are more aligned with the generator’s manifold, the model becomes more vulnerable to MIAs. Based on this insight, they propose AlignMI, a training-free method that enhances gradient-manifold alignment during the inversion process by explicitly projecting gradients onto the generator’s tangent space. Extensive experiments across diverse architectures validate the effectiveness of the proposed method.

**Questions:**

1. **Can you provide controlled experiments or causal analysis** to support the claim that gradient–manifold alignment directly leads to increased vulnerability to MIAs, beyond observed correlation?
2. **Why is the explicit projection (GMP) necessary**, given that the backpropagation path through the generator already restricts updates to the manifold span? Under what conditions does GMP outperform the implicit projection already present in GMI?
3. **Have you analyzed Jacobian degeneracy** in the generator across different latent regions? In cases where JG is rank-deficient, does the projection still help or even degrade the attack?

**Ethical Concerns:**

["NO or VERY MINOR ethics concerns only"]

**Final Justification:**

I have read the rebuttal and the comments of the other reviewers. My final rating is accept.

**Limitations:**

Please see my comments above.

**Paper Formatting Concerns:**

No major formatting issues.

**Quality:**

4

**Strengths And Weaknesses:**

#### **Strengths**

- The paper is well-written and easy to follow, with clear structure and logical progression.
- The proposed method achieves consistent and interpretable improvements over prior baselines in the MIA literature.
- The geometric perspective—analyzing the alignment between input gradients and the data manifold—is novel and intellectually stimulating.
- The observation that the generator structure inherently suppresses non-semantic ("off-manifold") gradient directions is insightful and valuable.

#### **Weaknesses**

- **Lack of causal analysis between alignment and leakage**:
  The central claim that "better alignment between gradients and the data manifold leads to higher vulnerability to MIAs" is only empirically correlated. No controlled experiments or ablation studies are provided to demonstrate a causal link.
- **Unclear motivation for explicit projection in Section 5**:
  In Section 3, it is claimed that the generator’s structure naturally imposes an implicit projection (e.g., $\Delta x = J_G J_G^\top \nabla_x L$ . However, Section 5 adds an explicit projection step without fully explaining why it provides further benefit over the implicit process.
- **No discussion of Jacobian degeneracy in practice**:
  The generator Jacobian is assumed to be full-rank, but in complex generators like StyleGAN, the Jacobian may degrade or collapse in certain regions of the latent s

---

> ### Author Rebuttal · Authors · 2025-07-31
>
> Thank you for your constructive comments and generous support! Please see our detailed responses to your comments and suggestions below.
>
> > W1. Lack of causal analysis between alignment and leakage: The central claim that "better alignment between gradients and the data manifold leads to higher vulnerability to MIAs" is only empirically correlated. No controlled experiments or ablation studies are provided to demonstrate a causal link.
>
> Thank you for your comment. We acknowledge the importance of establishing a causal relationship between gradient-manifold alignment and MIA vulnerability. In Section 6.2, we provide empirical validation of our hypothesis. The interpretation of our hypothesis validation requires a nuanced understanding, and we would like to clarify our reasoning through a step-by-step analysis:
>
> **1. Increase in $AS_{tr}$ leads to decrease in test accuracy (Figure 4(a)):**
>
> Ideally, to rigorously test the hypothesis, we would compare models with varying training-time alignment scores ($AS_{tr}$) while keeping test accuracy constant. However, in practice, alignment-aware training **inevitably reduces test accuracy**. We hypothesize that this trade-off could stem from inherent limitations of modern deep neural network architectures or the inductive biases introduced by SGD optimization. A parallel example is **adversarial robustness training** [r1], which improves robustness but consistently compromises natural accuracy.
>
> **2. Lower test accuracy leads to reduced model inversion vulnerability:**
>
> Previous work [r2] has shown that models with higher test accuracy tend to be more susceptible to generative MIAs, whereas those with lower test accuracy are typically more resistant. This is intuitively understandable—lower test accuracy implies weaker dependencies between input features and predictions, thus making it harder to infer input features from model outputs.
>
> **3. Interpreting Figure 5 (the inverted‑U‑shaped curve):**
>
> If gradient–manifold alignment were entirely unrelated to model inversion vulnerability, one would expect a **monotonic decline** in attack accuracy with increasing $AS_{tr}$, due to the corresponding drop in test accuracy (as discussed in Point 1). However, Figure 5 reveals a **non-monotonic** trend—attack accuracy initially increases before eventually declining. This is because improvements in gradient–manifold alignment **create a new attack surface**, leading to **increased model inversion vulnerability**. At early stages, the benefits of improved alignment outweigh the negative impact of reduced test accuracy, hence attack accuracy rises. Beyond a certain point, however, adverse effects of declining test accuracy become dominant, and attack accuracy begins to decline. This trend holds consistently across both architectures we studied. Therefore, we can infer that, for models with comparable test accuracy, those with better gradient–manifold alignment are more vulnerable to MIAs, thus supporting our core hypothesis.
>
>
>
> Additionally, the effectiveness of AlignMI compared to the baseline further supports the hypothesis, as it explicitly promotes better alignment and improves inversion performance.
>
>
> > W2. Unclear motivation for explicit projection in Section 5.
>
> Thank you for your comment. To clarify, we do not introduce an explicit projection operation in Section 5. Instead, we propose AlignMI, a method that constructs an alignment-enhanced loss gradient:
> $$
> \widetilde{\nabla}\mathcal{L}(\mathbf{x}) = \mathbb{E}_{\mathbf{x}'\sim p(\cdot\mid \mathbf{x})}\bigl[\nabla \mathcal{L}(\mathbf{x}')\bigr]
> $$
>
> This method is designed to better align with the generator manifold compared to the original loss gradient $\nabla_{\mathbf{x}} \mathcal{L}$. Unlike implicit projection, which naturally arises from the generator's structure during optimization, AlignMI explicitly modifies the gradient to enhance this alignment and leverages it to boost the effectiveness of generative model inversion attacks. This design offers **a complementary benefit** to the implicit mechanism described in Section 3 by directly reinforcing alignment, rather than relying solely on the generator's inductive bias.
>
> We support the AlignMI method with both quantitative and qualitative evidence: (1) Figure 12 (Appendix E.2) shows that AlignMI achieves higher alignment scores than the baseline. (2) Figures 13, 14, and 15 (Appendix E.4) visualize loss gradients and demonstrate that AlignMI produces gradients with more semantically meaningful features, indicating stronger alignment with the generator manifold.
>
>
> > W3. No discussion of Jacobian degeneracy in practice: The generator Jacobian is assumed to be full-rank, but in complex generators like StyleGAN, the Jacobian may degrade or collapse in certain regions of the latent space.
>
> Thank you for raising this important point. We acknowledge that the analysis in Section 3 assumes the generator Jacobian $J_G$ is full-rank, which forms the basis for interpreting $Range(J_G)$ as the tangent space of the generator manifold. This assumption enables a clean and tractable theoretical framework for analyzing the alignment between gradients and the generator’s manifold.
>
> While Jacobian degeneracy is theoretically possible, the likelihood of such degeneracy is generally low in practice. This is primarily due to the dimensional disparity between the image space (e.g., 64×64×3 or 224×224×3) and the latent space (e.g., dimension 512), which reduces the risk of rank deficiency in the Jacobian.
>
>
> To further investigate, we conducted a statistical analysis of the Jacobian’s rank during the inversion optimization process. Specifically, we employed singular value decomposition (SVD) to evaluate the smallest singular values of $J_G$, as a non-zero minimum singular value indicates full rank. We sampled **10,000 samples** during the optimization and report the distribution of the smallest singular values in the table below.
>
>
> | Smallest Singular Value Range | Percentage of Data Points (%) |
> |--------------------|-------------------------------|
> | 0.001 - 0.01       | 0.0%                          |
> | 0.01 - 0.1         | 99.3%                         |
> | 0.1 - 1            | 0.7%                          |
>
> The results indicate that the generator Jacobian remains consistently full-rank throughout the inversion process, as evidenced by the strictly positive smallest singular values observed across all sampled instances.
>
> **We hope the clarifications provided have addressed your concerns and reinforced your confidence in our work. Please feel free to reach out if you have any further questions or require additional details.**
>
> ---
> **References:**
>
> [r1] Fredrikson et al. Model inversion attacks that exploit confidence information and basic countermeasures. In CCS, 2015.
>
> [r2] Zhang et al. The secret revealer: Generative model-inversion attacks against deep neural networks. In CVPR, 2020.

---

> > ### Comment · Reviewer_WVQ7 · 2025-08-05
> >
> > Thank you for your detailed and thoughtful responses to my comments. I appreciate the effort to clarify the reasoning behind the motivation for AlignMI, and the practical considerations regarding Jacobian degeneracy. I will maintain my score as well as my judgment of acceptance.

---

> ### Author Response · Authors · 2025-08-05
>
> Dear Reviewer WVQ7,
>
> We appreciate your valuable feedback and generous support, as well as the time you dedicated to reviewing our work.
>
> Sincerely,
>
> The Anonymous Authors

---

### Official Review · Reviewer_psS7 · 2025-07-01

**Clarity:** 3
**Significance:** 4
**Originality:** 3
**Rating:** 5
**Confidence:** 3

**Summary:**

This paper studies the problem of generative model inversion, which loosely consists of retrieving class-representative data samples from a trained classifier using a GAN model. First, the authors observe that during the model inversion phase, backpropagation of the inversion-time loss through the generator acts as a projection of the loss onto the tangent space of the manifold where the data distribution lies. However, they empirically observe that the loss gradients remain poorly aligned with the tangent space during model inversion, suggesting that the gradients are noisy and thus carry limited semantically meaningful information for guiding the inversion process. The authors hypothesize that models are more vulnerable to inversion attacks when their inversion-time loss gradients are well aligned with the data manifold. To test this hypothesis, they propose two methods to improve the manifold alignment of the loss gradients: (1) fine-tuning the models by adding a regularization term that promotes alignment with the manifold, and (2) averaging multiple gradients sampled from a small neighborhood during the inversion phase. Finally, the authors empirically validate their hypothesis with experiments using two standard classifiers on the CelebA dataset.

**Questions:**

- 1/ how do you explain the behavior of the VGG16 model in Figure 5?
- 2/ Could a similar technique, based on your analysis, be developed to reduce the MIA vulnerability of models?

**Ethical Concerns:**

["NO or VERY MINOR ethics concerns only"]

**Final Justification:**

Authors adressed my main concern (W2) and added a new experiments relatively to (Q2).

**Limitations:**

yes. However, please put this discussion on the limitations of the method in the main and not in the appendix.

**Quality:**

3

**Strengths And Weaknesses:**

As someone not familiar with model inversion, I enjoyed reading this paper. The approach seems new as far as I can tell.

Strengths:

- The paper establishes a link with MIA vulnerability and loss gradient alignment with the data manifold
- The authors have designed a training-free method to increase the manifold-alignment and the empirically show in Tab 2 that this method is effective to increase MIA vulnerability.

Weaknesses:

- Apparently, the method is limited to relatively low-dimensional setting due to its computational cost
- In my opinion, the experiment section doesn’t completely validate the author’s hypothesis formulated in section 4. Indeed, the authors observe that the penalized training approach inevitably deteriorates the performances of the classifier, which has apparently itself an impact on MIA vulnerability. I’m therefore not sure that this approach can be used to rigorously validate the authors hypothesis.  At the opposite, the training-free approach seems to me to better validate the author’s hypothesis but it would have been nice to give also the evolution of the alignments scores because its not completely clear if this latter approach effectively improves the manifold-alignment scores since its mostly based on an intuition.

---

> ### Author Rebuttal · Authors · 2025-07-31
>
> We sincerely thank you for your constructive comments and generous support! Please see our detailed responses to your comments and suggestions below.
>
> > W1. The method is limited to relatively low-dimensional setting due to its computational cost.
>
> Thank you for the observation. The low-dimensional setting is used **only for hypothesis validation**, as computing and storing the decoder Jacobian at high resolution (e.g., 224×224×3) would be prohibitively expensive. Since the goal here is to validate the main hypothesis rather than implement a practical attack, experiments in low-dimensional settings are sufficient and appropriate for empirical analysis. Importantly, when evaluating the **AlignMI attack algorithm**, we conduct experiments in **both low- and high-resolution settings** to demonstrate its effectiveness in realistic scenarios.
>
>
> > W2.1. In my opinion, the experiment section doesn’t completely validate the author’s hypothesis formulated in section 4 ... I’m therefore not sure that this approach can be used to rigorously validate the authors hypothesis.
>
> > Q1. How do you explain the behavior of the VGG16 model in Figure 5?
>
> Thank you for raising this important point. We agree that the interpretation of our hypothesis validation requires a nuanced understanding, and we would like to clarify our reasoning through a step-by-step analysis:
>
> **1. Increase in $AS_{tr}$ leads to decrease in test accuracy (Figure 4(a)):**
>
> Ideally, to rigorously test the hypothesis, we would compare models with varying training-time alignment scores ($AS_{tr}$) while keeping test accuracy constant. However, in practice, alignment-aware training **inevitably reduces test accuracy**. We hypothesize that this trade-off could stem from inherent limitations of modern deep neural network architectures or the inductive biases introduced by SGD optimization. A parallel example is **adversarial robustness training** [r1], which improves robustness but consistently compromises natural accuracy.
>
> **2. Lower test accuracy leads to reduced model inversion vulnerability:**
>
> Previous work [r2] has shown that models with higher test accuracy tend to be more susceptible to generative MIAs, whereas those with lower test accuracy are typically more resistant. This is intuitively understandable—lower test accuracy implies weaker dependencies between input features and predictions, thus making it harder to infer input features from model outputs.
>
> **3. Interpreting Figure 5 (the inverted-U-shaped curve):**
>
> If gradient–manifold alignment were entirely unrelated to model inversion vulnerability, one would expect a **monotonic decline** in attack accuracy with increasing $AS_{tr}$, due to the corresponding drop in test accuracy (as discussed in Point 1). However, Figure 5 reveals a **non-monotonic** trend—attack accuracy initially increases before eventually declining. This is because improvements in gradient–manifold alignment **create a new attack surface**, leading to **increased model inversion vulnerability**. At early stages, the benefits of improved alignment outweigh the negative impact of reduced test accuracy, hence attack accuracy rises. Beyond a certain point, however, adverse effects of declining test accuracy become dominant, and attack accuracy begins to decline. This trend holds consistently across both architectures we studied. Therefore, we can infer that, for models with comparable test accuracy, those with better gradient–manifold alignment are more vulnerable to MIAs, thus supporting our core hypothesis.
>
>
> > W2.2. At the opposite, the training-free approach seems to me to better validate the author’s hypothesis ... since it's mostly based on an intuition.
>
> Thank you for this insightful observation. As you noted, the training-free approach is primarily grounded in geometric intuition, whereas the alignment-aware training provides a more principled way to validate the hypothesis, as discussed earlier.
>
> To address your request for more concrete evidence, we note that our initial submission already included quantitative and qualitative evidence demonstrating that the training-free AlignMI approach indeed improves gradient–manifold alignment. Specifically, **Figure 12** in Appendix E.2  presents a comparison of the $AS_{tr}$ distributions between the baseline and both the PAA and TAA variants of AlignMI. The results clearly show that the AlignMI approach increases the alignment scores relative to the baseline.
>
> Furthermore, **Figures 13, 14, and 15** in Appendix E.4 visualize the loss gradients for both the baseline and AlignMI. These visualizations reveal that the gradients produced by AlignMI exhibit more semantically meaningful features, offering qualitative evidence that our approach enhances alignment between the loss gradients and the generator manifold.
>
> > Q2. Could a similar technique, based on your analysis, be developed to reduce the MIA vulnerability of models?
>
> We appreciate your insightful question and your intuition is indeed correct. We investigated how reducing $AS_{tr}$ (i.e., gradient-manifold alignment) affects a model’s vulnerability to generative MIAs. As shown in the following table, decreasing this alignment consistently leads to lower attack success rates, indicating a reduction in vulnerability.
>
>
> | Target Model | Training Variant | $AS_{tr}$ | Test Acc $\uparrow$ | Acc@1 $\downarrow$ | KNN Dist $\uparrow$ |
> |--------------|------------|-----------|----------|-------|-----------|
> | VGG16        | Model A    | 0.1655    | 93.00    | 88.83 | 1162.49   |
> |              | Model B    | 0.1034    | 92.97    | 85.06 | 1213.12   |
> |              | Model C    | 0.0470    | 93.75    | 75.56 | 1326.68   |
> | IR152        | Model A    | 0.2219    | 96.88    | 85.87 | 1300.92   |
> |              | Model B    | 0.1558    | 96.48    | 84.70 | 1320.47   |
> |              | Model C    | 0.1062    | 96.48    | 81.70 | 1350.10   |
>
> These findings support the promising direction that techniques explicitly designed to reduce gradient–manifold alignment could be developed as effective defenses against generative MIAs. We view this as a compelling area for future work and appreciate your suggestion in helping shape that research direction.
>
>
>
> > Re: Discussion of Limitations.
>
> Thank you for pointing this out. Due to space constraints in the initial submission, we included the discussion on the limitations of our method in the appendix. We agree that this discussion is important and will move it to the main text in the final version of the paper to ensure greater clarity and completeness.
>
>
> **We hope the clarifications provided have addressed your concerns and reinforced your confidence in our work. Please feel free to reach out if you have any further questions or require additional details.**
>
>
> ---
> **References:**
>
> [r1] Fredrikson et al. Model inversion attacks that exploit confidence information and basic countermeasures. In CCS, 2015.
>
> [r2] Zhang et al. The secret revealer: Generative model-inversion attacks against deep neural networks. In CVPR, 2020.

---

> > ### Comment · Reviewer_psS7 · 2025-08-01
> >
> > Thank you for answering my questions and adressing my concerns. I updated my rating.

---

> > > ### Author Response · Authors · 2025-08-01
> > >
> > > Dear Reviewer psS7,
> > >
> > > We sincerely thank you once again for the time and effort you dedicated to reviewing our paper, as well as for your generous support. Your insightful comments have been very valuable in improving our work.
> > >
> > > Sincerely,
> > >
> > > The Anonymous Authors

---

### Official Review · Reviewer_Xy65 · 2025-07-02

**Clarity:** 3
**Significance:** 3
**Originality:** 3
**Rating:** 5
**Confidence:** 3

**Summary:**

This paper studies the problem of model inversion attacks (MIA). The authors find that the gradients of inversion loss are noisy and generative inversion implicitly denoises them by projecting onto the tangent space of data manifold. Since alignment of gradient and the tangent space makes models more vulnerable to MIA, the authors propose a training objective too promote it. They further propose a training-free approach to enhance gradient–manifold alignment during inversion.

**Questions:**

(derived from "weaknesses")
1. Is it possible to show experiments from multiple runs for statistical significance?
2. Why is the alignment phenomenon so specific to the MIA problem?
3. Can there be data leakage from the use of VAE?

**Ethical Concerns:**

["NO or VERY MINOR ethics concerns only"]

**Final Justification:**

I thank the authors for the rebuttal. I keep the original score since it is already positive.

**Limitations:**

yes

**Quality:**

3

**Strengths And Weaknesses:**

Strengths:
- The hypothesis about alignment provides novel insights for the MIA problem. I believe it will be interesting to a large audience.
- The hypothesis is validated by extensive experiments. The proposed method also consistently improve the performance.
- The paper is well-written. The problem is well motivated. The methodology is clearly presented. The experiments are reproducible.

Weaknesses:
1. In checklist #7, the authors acknowledge the lack of statistical significance. Although they mentioned that this is due to computational cost, it may still reduce reliability.
2. The intuition is not fully convincing. Why is the alignment phenomenon so specific to the MIA problem? It seems that in any tasks involving optimization over image space, having gradient along data manifold can be helpful.
3. The use of VAE may be unfair. The method relies on a pretrained VAE to estimate the data manifold. Then, it means that there is already some understanding of the data before implementing the attack.

---

> ### Author Rebuttal · Authors · 2025-07-31
>
> Thank you for your thoughtful comments and generous support! We appreciate your recognition of our contributions and the strengths of the paper. Please see our detailed responses to your comments and suggestions below.
>
> > W1, Q1. Lack of multiple experimental runs to ensure statistical significance.
>
> Thank you for pointing this out. To address this concern, we have conducted three independent runs (with different random seeds) of our key experiments in Table 2 to assess the consistency and statistical significance of our results. We report the mean and standard deviation of the results in the table below:
>
>
> |              |        | **CelebA**        |                   | **FaceScrub**     |                   |
> |--------------|--------|-------------------|-------------------|-------------------|-------------------|
> | **Target Model** | Method | Acc@1 $\uparrow$  | KNN Dist $\downarrow$ | Acc@1 $\uparrow$  | KNN Dist $\downarrow$ |
> | **ResNet‑18**    | PPA    | 86.04 $\pm$ 0.04  | 0.6897 $\pm$ 0.0002 | 81.86 $\pm$ 0.35  | 0.7943 $\pm$ 0.0024 |
> |              | + PAA (ours) | 88.69 $\pm$ 0.28  | 0.6691 $\pm$ 0.0011 | 83.98 $\pm$ 0.22  | 0.7761 $\pm$ 0.0025 |
> |              | + TAA (ours) | 90.17 $\pm$ 1.15  | 0.6624 $\pm$ 0.0001 | 93.46 $\pm$ 0.30  | 0.6984 $\pm$ 0.0071 |
> | **DenseNet‑121** | PPA    | 81.15 $\pm$ 0.80  | 0.7148 $\pm$ 0.0054 | 74.52 $\pm$ 1.78  | 0.8007 $\pm$ 0.0178 |
> |              | + PAA (ours) | 85.42 $\pm$ 0.23  | 0.6870 $\pm$ 0.0005 | 80.35 $\pm$ 0.06  | 0.7363 $\pm$ 0.0021 |
> |              | + TAA (ours) | 87.84 $\pm$ 0.73  | 0.6832 $\pm$ 0.0095 | 84.91 $\pm$ 0.14  | 0.7287 $\pm$ 0.0036 |
> | **ResNeSt‑50**   | PPA    | 71.16 $\pm$ 0.10  | 0.7926 $\pm$ 0.0004 | 71.62 $\pm$ 0.20  | 0.8315 $\pm$ 0.0002 |
> |              | + PAA (ours) | 76.08 $\pm$ 0.17  | 0.7614 $\pm$ 0.0012 | 73.52 $\pm$ 0.55  | 0.8074 $\pm$ 0.0042 |
> |              | + TAA (ours) | 78.07 $\pm$ 1.41  | 0.7665 $\pm$ 0.0120 | 84.10 $\pm$ 0.04 | 0.7568 $\pm$ 0.0005 |
>
>
>
> These results confirm that our method consistently outperforms the baselines across runs, demonstrating the statistical robustness of our approach. We will include the updated results in the main results section (Section 6.3) of the final version of our paper to enhance the reliability of the proposed method.
>
>
>
>
> > W2, Q2. The intuition is not fully convincing ... having gradient along data manifold can be helpful.
>
> Thank you for the thoughtful question. The motivation behind our work is grounded in the evolution of model inversion attacks (MIAs). The seminal work [r1] formulated MIAs as an input-space optimization problem. However, this traditional approach often produces noisy features, especially when the target DNNs are trained on high-dimensional data.
>
> To address this, generative model inversion (GMI) approaches [r2] introduced image priors via pretrained generators (e.g., GANs), constraining the optimization to the generator’s latent space. This significantly improved visual quality and semantic relevance of reconstructed samples. These advancements naturally led us to ask: Why are GMI approaches so effective?
>
> This question drove the analysis presented in Section 3. Our derivation shows that GMI methods implicitly denoise gradients by projecting them onto the generator’s manifold, thereby preserving only the gradient components aligned with the tangent space of the generator manifold and filtering out directions that deviate from it (Figure 2). Surprisingly, despite the effectiveness of GMI approaches, we found the degree of alignment between the gradients and the generator manifold to be **remarkably low** (Figure 3).
>
> This observation led us to our core idea: can we explicitly **promote gradient–manifold alignment** to further improve inversion performance? To the best of our knowledge, this question has not been explored in prior work, and it forms one of the central contributions of our paper. Before proposing a new attack method, we rigorously validated our hypothesis that improving gradient–manifold alignment would boost inversion performance. This hypothesis is formalized in Section 4 and empirically validated in Section 6.2.
>
>
> > W3, Q3. The use of VAE may be unfair ... understanding of the data before implementing the attack.
>
> We would like to clarify that the use of VAE is not **intended to facilitate the attack itself**, but rather to estimate the data manifold for validating our hypothesis. In this context, model inversion attacks are used as an **evaluation tool** to assess how different levels of gradient–manifold alignment impact a model’s inversion robustness. Therefore, the use of a pretrained VAE to estimate the data manifold is not meant to simulate a realistic attack scenario, but instead to **enable a principled evaluation** of the core hypothesis being examined.
>
> **We hope the clarifications provided have addressed your concerns and reinforced your confidence in our work. Please feel free to reach out if you have any further questions or require additional details.**
>
> ---
> **References:**
>
> [r1] Fredrikson et al. Model inversion attacks that exploit confidence information and basic countermeasures. In CCS, 2015.
>
> [r2] Zhang et al. The secret revealer: Generative model-inversion attacks against deep neural networks. In CVPR, 2020.

---

> > ### Comment · Reviewer_Xy65 · 2025-08-05
> >
> > Thank you for your response. I keep my rating of this paper.

---

> > > ### Author Response · Authors · 2025-08-05
> > >
> > > Dear Reviewer Xy65,
> > >
> > > We sincerely thank you again for your time, valuable feedback, and generous support of our work.
> > >
> > > Sincerely,
> > >
> > > The Anonymous Authors

---

### Note · Authors · 2025-08-13

We sincerely thank all reviewers for their constructive feedback, positive evaluations, and recognition of our contributions, and we are grateful to the Area Chair for careful coordination and guidance throughout the review process. Reviewers highlighted the novelty of our geometric perspective on gradient–manifold alignment for understanding the vulnerability of model inversion attacks (MIAs) (Xy65, psS7, WVQ7). They agreed that our hypothesis is well supported by extensive experiments (Xy65), that the AlignMI approach consistently yields improvements over baseline methods (Xy65, WVQ7, 6LmF), and that our methodology is clearly presented and the paper well written (Xy65, psS7, WVQ7).

We appreciate the reviewers for raising points that helped strengthen and clarify our work. In particular, we have addressed: the need for statistical significance through multiple runs (Xy65, 6LmF); the importance of alignment specificity for MIAs (Xy65); the role of a pretrained VAE in validating our hypothesis (Xy65); a clearer explanation of how our experiments verify the hypothesis (psS7, WVQ7); exploration of defenses informed by our analysis (psS7); Jacobian degeneracy analysis (WVQ7); and distinctions between MIAs and related inversion tasks (6LmF).

In the final version, we will incorporate the reviewers’ feedback by including updated multi-run results with implementation details, and refining explanations of our hypothesis validation. We thank the reviewers for acknowledging the novelty, clarity, and potential impact of our work, and we look forward to sharing the improved version with the community.

---

### Decision · Program_Chairs · 2025-09-17

**Decision:**

Accept (poster)

**Comment:**

In this paper, the authors study Model Inversion Attacks (MIAs), where one seeks class-representative samples from a trained generative model, which poses potential privacy and security risks for trained models.  The authors make an original observation that MIAs implicitly denoises inversion loss gradients by projecting them onto the generators signal manifold.  As a result, they hypothesize that models are more vulnerable to MIAs when their loss gradients align with tangent spaces to the generative manifold.  This leads the authors to introduce an MIA that works by enhancing alignment with the gradient manifold. The resulting MIA outperforms state-of-the-art baselines.  The conclusions in the paper are original and provide conceptual understanding of ML systems that leads to new algorithmic approaches.  It is likely to be of interest to a broad community.